# Impact of Alcohol on Occupational Health and Safety in the Construction Industry at Workplaces with Scaffoldings

**Marek Sawicki and Mariusz Szóstak \*** 

Department of Building Engineering, Faculty of Civil Engineering, Wroclaw University of Science and Technology, 50-370 Wrocław, Poland; marek.sawicki@pwr.edu.pl
**\*** Correspondence: mariusz.szostak@pwr.edu.pl

**Abstract:** The value, care, and customs of workers are essential in terms of occupational health and safety. The abuse of alcohol is widely regarded as a serious threat to the lives, health, and safety of employees. The aim of the research was to identify the main problems that are associated with alcohol abuse and consumption at work among employees in the construction industry, with particular emphasis on workstations where work is carried out on construction scaffoldings. Data for the analysis were obtained from two different sources. The first one was post-accident documentation on occupational accidents. The second one was surveys collected during the research project. This study confirmed that excessive and disproportionate alcohol consumption can be the cause of an accident, and consequently death at workplaces with scaffolding. Of 219 accident reports, 17.4% indicated alcohol as a contributing factor. Analysis of accident documentations shows that in cases where alcohol was indicated as a contributing factor in an accident, the alcohol was consumed during the workday. The results obtained on the basis of the conducted research were able to constitute a justification for the directions of preventive actions carried out in order to reduce the number of occupational accidents in the construction industry caused by alcohol.

**Keywords:** health and safety; workplace; construction industry; scaffoldings; alcohol

## 1. Introduction

Alcohol abuse is widely considered a serious threat to the life, health, and safety of employees and bystanders [1]. The harmful use of alcohol is one of the leading risk factors for population health and cause of death worldwide [2]. According to a World Health Organization report, alcohol is in the third place among the risk factors for human health, and over 60 types of diseases and injuries are associated with it [3,4]. Alcohol consumption disorder (alcohol addiction) is a serious illness with far-reaching consequences for life and health [5].

It is well known that use of drugs (alcohol, cigarettes, psychoactive substances) influences the occurrence of an accident situation in various branches of the economy, as well as in the construction industry [6]. According to scientific research on alcohol abuse, construction workers are at the forefront of professions in which the percentage of alcoholics is significant [7,8].

A construction worker is a typically male teamwork profession that requires physical effort [9], is performed outdoors, and is often carried out in difficult weather conditions [10]. These conditions are conducive to drinking alcohol for relaxation [11]. Drinking alcohol is an accepted social and cultural habit in most Western countries [12]. An additional factor contributing to alcohol abuse among construction workers is the mental tension caused by working in the construction industry, which involves stress, time pressure, and superiors, as well as depression [13]. Alcohol abuse also has

negative social consequences [14]. Alcohol consumption has a negative impact on human health [15]. Moreover, accidents caused by alcohol consumption translate into economic consequences [16], which cause, among other factors, a decline in the well-being of societies [17], as well as an increase in the direct cost of healthcare [18].

The phenomenon of alcohol abuse and consumption at work is a common problem in many sectors of the economy, especially in the construction industry [19]. Many construction workers have alcohol-related problems [20,21]. This problem also applies to workers at workplaces with scaffoldings [22]. Alcohol increases the risk of accident situations. Moreover, the most common events that cause accidents to workers on scaffolding are falls from height caused by reduced concentration, among other factors [23].

Statistical data published by the Statistical Office of the European Union (Eurostat) are the basis for the statement that the serious problem of alcohol consumption exists [24]. Only 16.4% of men in 2014 declared that they had "never or not in the last 12 months" consumed alcohol. This value is the frequency of alcohol consumption for 27 countries of the European Union (EU27). On the other hand, consumption of alcohol "every day" and "every week" were declared by 14.7% and 35.5%, respectively, of men in the 27 countries of the EU. The highest daily frequency of alcohol consumption was in Portugal (38.6%), and the highest weekly frequency in the United Kingdom (51.6%).

Furthermore, statistical data published by the Central Statistical Office of Poland [25] regarding the average annual alcohol consumption per capita of a statistical citizen is worrying. According to these data, the level of alcohol consumption increased from approximately 6 L per person per year in 2002 to 10 L in 2018. The rise in the amount of alcohol consumption resulted in a decrease in the rate of life expectancy, and from 2013 halted the tendency towards increased life expectancy. In addition, according to data in Poland, alcoholics constitute about 2% of the population, i.e., 600,000–800,000 people, and alcohol abusers constitute about 12% of the population, i.e., 3.6–4.8 million people.

There were two aims in this study. First, the overarching goal of the research was to identify the main problems associated with the consumption of alcohol at work among employees in the construction industry, with particular emphasis on jobs related to work on scaffolding. The second goal of the study was to determine patterns of alcohol consumption among construction workers in Poland.

This research is relevant to the development of interventions for reduction of occupational accidents and of interest to public health. The results obtained on the basis of the conducted research may be a justification for the directions of preventive actions carried out in order to reduce the number of occupational accidents in the construction industry caused by alcohol. This will significantly contribute to an increase of the level of occupational safety in the construction industry.

## 2. Literature Review

The results of studies conducted by Benner's team concerning the relationship between the amount of consumed alcohol and the general cause of death of construction workers are important with regards to the analyzed issue [26]. The study was conducted on 8043 German construction workers aged 25–64 who were employed as the following professions: plumber, carpenter, painter, plasterer, bricklayer, unskilled construction worker, office worker, engineer, and architect. Information regarding smoking, the number of people consuming alcohol, and the amount of alcohol consumed daily was obtained on the basis of medical investigations carried out by medical staff. Thus, only 7.4% of all people participating in the study declared abstinence, while the remaining people consumed alcohol daily in an amount from 1 g/day to over 100 g/day. The research assumed that 50 g of alcohol per day corresponds to the consumption of 1 L of beer or 0.5 L of wine, while 100 g of alcohol per day corresponds to 2 L of beer or 1 L of wine. Occasional drinkers (from 1–49 g/day) were the most numerous group, constituting over half of the total drinkers (53.3%). A total of 26.4% of employees declared alcohol consumption at an amount of 50–99 g/day, while 12.9% declared the consumption of over 100 g/day. In addition, research showed that alcohol consumption is highly correlated with smoking. Over 65.0% of respondents smoked cigarettes. Hypertension was the most common issue

among workers who smoked and drank. During the study, there were 172 deaths—mainly in the group of employees declaring consumption of an amount of 1–49 g/day. Studies showed that the highest number of deaths occurred among people who declared alcohol consumption of an amount of over 50 g/day.

In turn, the Australian research team headed by Prof. H. Biggs, in cooperation with a large construction company and with the support of the Research Center of the National Fund for Sustainable Construction, implemented a research project that aimed to determine the impact of alcohol and other drugs on safety in the construction industry [27]. In total, 494 people were surveyed, with an average age of 35.7 (± 11.4). In the conducted surveys, employees were asked about the amount and frequency of alcohol consumption, drinking behavior that could indicate alcohol addiction, as well as the negative consequences of alcohol consumption. The obtained results showed that 58% of the surveyed people (i.e., 286 people) had contact with alcohol in their daily work. A high risk of alcohol-related problems was identified among 185 respondents, while alcohol addiction occurred in 43 respondents. The research was of great importance for Australia, and the results of the research were used to introduce harmonized standards in the construction industry in the field of occupational health and safety.

Subsequent studies conducted among young construction workers (the average age of people participating in the study was 21 years old) showed that about 65% of the respondents performed harmful and dangerous practices related to drinking alcohol. A total of 39% of respondents consumed alcohol 2–3 times a week, while 36% indicated that they consumed 10 or more alcoholic beverages at one time. The research also identified the positive correlation between harmful employee behavior and violence among drinkers (verbal violence, racial harassment, threats, attacks of aggression) [28].

According to the literature survey, the phenomenon of alcohol abuse and consumption at work is a common problem in many sectors of the economy [29]. This problem especially occurs in male-dominated industries [30]. On the basis of the Australian and New Zealand standard classification of economic sectors, industry is defined as male-dominated, with men accounting for around 70% of employees [31]. For example, in Australia, such sectors of the economy are agriculture (70%), construction (88%), mining (82%), production (74%), transport (77%) and municipal services (76%). In turn, in Poland, the 70% criterion is met by sectors of the economy such as mining and extraction (90%), construction (89%), energy production and supply (79%), transport and storage (77%), and municipal services (76%) [32]. Therefore, it can be concluded that the construction industry is a sector dominated by men who often have a problem with alcohol abuse and consumption at work.

The consumption of alcohol at a workstation leads to the deterioration of employee performance, which can in turn result in employee absenteeism at work, an occupational accident, other significant problems that are related to the occupational safety of employees, or interpersonal problems between workers [33]. It has also been reported that alcohol increases the risk of aggression and violence towards colleagues, as well as towards family members [34].

The increased risk of an accident under the influence of alcohol results from, among others, a reduced attention and cognitive ability, as well as reduced concentration and delay in making decisions and taking actions that can prevent the occurrence of an accident [35].

Alcohol influences, among others, psychomotor dysfunction (disorders of balance, speaking, thinking, and concentration; impaired motor coordination; reduced level of perception of threats; and others), and also damages many organs. The risk of getting sick and dying from alcohol abuse increases with an increasing alcohol consumption [36]. Alcohol consumption is associated with an increased risk of cardiovascular disease, elevated blood pressure, and liver disease [37]. Mental disorders such as anxiety and depression are common comorbidities in people with alcohol consumption disorders [38]. In addition, drinking large amounts of alcohol on an empty stomach may lead to hypoglycemia and a drop in blood sugar levels [39]. This condition can lead to a decrease in manual skills, fatigue, and irritability, which can in turn cause accidents. Often, people who consumed alcohol the previous day believe that they are already sober on the basis of their own judgment and

well-being, which is often not true. Even the prevailing belief among many people that the decrease in blood alcohol content can be accelerated by a long sleep, a cold shower, or drinking water and coffee, is not true.

In addition, a very common consequence of consuming an excessive amount of alcoholic beverages, which usually occurs several hours, or the next day, after the consumption, is the phenomenon of malaise—the so-called alcohol hangover [40]. The most common ailments accompanying this phenomenon are headache, thirst, photophobia, hypersensitivity to noise, nausea, problems with concentration, weakness, irritability, and a decrease in the employee's ability to perceive threats and respond to them [41]. An employee under the influence of alcohol and psychoactive substances becomes both a threat to himself and to other people [42]. Alcohol also reduces the experience of pain. For example, the average pain threshold for employees without alcohol consumption is lower by 23% than that of occasional drinkers, and 42% lower than those who drink daily [43].

An investigation carried out by Liu et al. [44] determined that the average human body is able to burn 0.12–0.15‰ of alcohol per hour. The process of alcohol burning depends on many factors, among others, including sex, body weight, individual predispositions associated with metabolism, the amount and type of consumed food, and the state of health of the body. Small amounts of consumed alcohol are excreted with exhaled air and urine, only larger amounts attack the body and reach the brain and other organs. Excessive and prolonged alcohol consumption can lead to alcoholic liver damage (liver cirrhosis), myocardial damage, and brain damage (ischemic stroke).

In order to prevent excessive alcohol consumption by citizens in many countries around the world, guidelines are set for "safe" alcohol consumption. There are significant differences in alcohol policy between individual European countries and regions. National alcohol policies in the European Union are constantly changing [45]. Guidelines on alcohol policy and alcohol consumption are being developed in individual countries. Guidelines are usually developed by governments, e.g., the Ministry of Health, other Ministries, or government agencies that are responsible for alcohol policy. Information on alcohol consumption is based on the results of biomedical research and the relationship between the dose of alcohol consumed and the individual response of the body. Guidelines referring to alcohol consumption are presented in two forms: as grams of ethyl alcohol, or as the amount of "standard" drinks consumed during a day. For the quantity of "standard" drinks, the number of grams of ethyl alcohol per unit volume is determined, which enables an easy conversion to the level of pure alcohol. For example, the Portuguese National Food and Nutrition Council provides in its recommendations on "standard" units on the basis of wine consumption, while the Romanian Ministry of Health distinguishes in its guidelines the level of alcohol consumed in beer and wine. In turn, the guidelines of the Australian Council on Health and Medical Research on alcohol provide patterns of alcohol consumption, which are associated with long-term (chronic), as well as short-term (acute) harm to people consuming alcohol. These guidelines determine the appropriate level of the risk of health and social problems, including injury and death [46].

In turn, in Poland, a "standard" dose of alcohol is equal to 10 g or 12.5 mL of pure ethyl alcohol. This dose, calculated for the most commonly consumed types of alcoholic beverages, is equal to 200 g of 4.5% beer, 100 g of 10.0% wine, and 25 g of 40.0% vodka. When considering that alcoholic beverages are usually sold in volume measures, alcohol values in typical commercial units amount to 22.5 mL/18 g of alcohol for 500 mL of 4.5% beer (a "large beer"), 21 mL/16.8 g alcohol for 175 mL of 10.0% wine (a glass), and 20 mL/16 g of alcohol for 50 mL of 40.0% vodka. A low health risk for men is considered to be the consumption of up to four standard doses of alcohol per day (125 mL of vodka or 0.4 L of wine or two "large beers"), no more than five times a week. It is considered risky to drink more than six standard portions of alcohol per day [47,48].

According to the International Center for Alcohol Policies (ICAP) 14 Report called "International recommendations on alcohol consumption" [49], and in accordance with the recommendations of the World Health Organization (WHO), it is suggested, in Poland, to consume alcohol in an amount not exceeding two units per day (i.e., 20 g), a maximum of five times a week (no more than 100 g per

week), with at least two days per week being without alcohol. According to the report, the smallest recommended amount of consumed alcohol, existing in Japan, is equal to one daily unit—19.75 g according to the Ministry of Health, Labor and Social Affairs. In contrast, the highest permitted amount of consumed alcohol occurs in France (up to five daily units, i.e., 60 g according to their National Medical Academy) and Spain (up to 70 g/day according to their Department of Health and Social Policy).

The particular goal of the study was to determine patterns of alcohol consumption among construction workers in Poland. In this study concerning alcohol consumption patterns, attention was paid to two important aspects that are related to the consumption of alcohol by employees: alcohol consumption in the workplace, i.e., during working hours or immediately before or after work, and also alcohol consumption after normal working hours. Although a relatively small number of people consume alcohol immediately before or during working hours, a significant proportion of employees consume alcohol in their free time after business hours [46]. Defining patterns is essential for the appropriate implementation of effective public health measures regarding prevention [50].

To date, there has not been any similar research carried out in Poland, and there is currently no information about alcohol consumption on Polish construction sites. Therefore, this paper is an attempt to fill this research gap.

## 3. Methodology of Research

The conducted research examined the consumption of alcohol by employees with regards to the daily amount of alcohol that was consumed by construction workers working at workplaces with the use of construction scaffolding. Data for the analysis were obtained from two sources. Figure 1 presents the map of the sites in Poland where the research was realized.

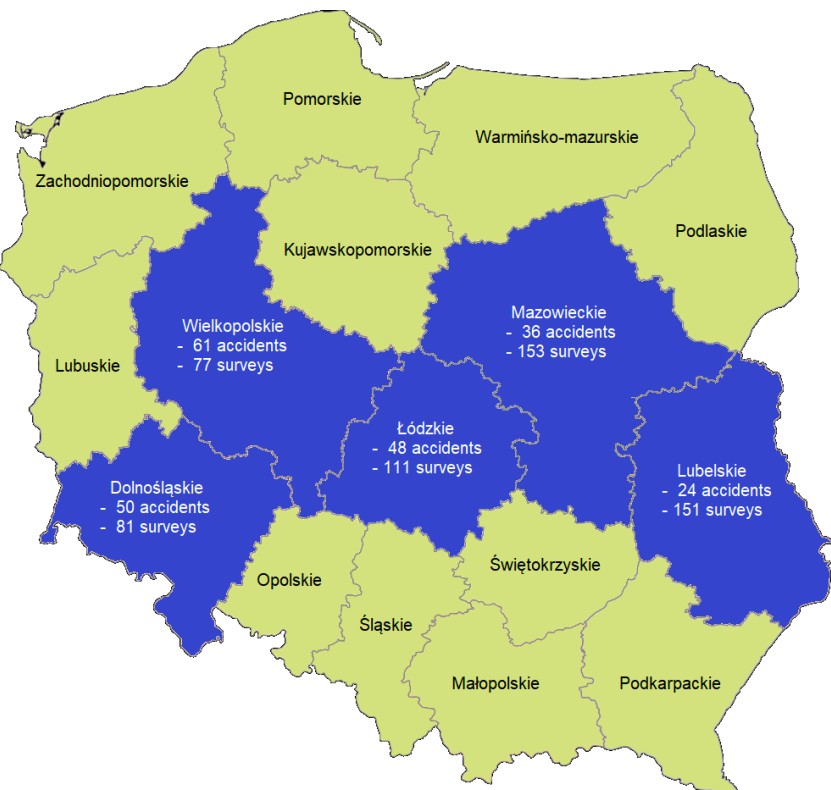

**Figure 1.** The map of the sites in Poland where the research was realized (own elaboration).

### 3.1. Accident Documentation

The first source of data concerning occupational accidents involved accident documentation (inspection reports) prepared by inspectors of the National Labor Inspectorate in Poland [23]. A total of 219 people who were injured in 2008–2017 in occupational accidents involving construction scaffolding were analyzed. On the basis of the Control Protocol, i.e., on the basis of a description of the circumstances and causes of the accident, it was possible to obtain information about the health status of an injured person during and after the accident. The report includes information, confirmed by a police officer or doctor admitting the injured person to the hospital ward, concerning blood alcohol content (in units: ‰ or mg/L); the consumed amount of alcoholic beverages; and also the statement "state after consumption", i.e., indicating alcohol consumption or "intoxication state". The amount of alcohol in the bloodstream is called blood alcohol concentration or BAC. BAC can be measured with a breathalyzer or by analyzing a sample of blood. It is measured by the number of grams of alcohol in 100 mL of blood. For example, a BAC of 0.08 means 0.08 g of alcohol in every 100 mL of blood.

It is worth noting that in Poland, as well as in many other European countries, alcohol consumption by employees is classified according to the level of ethyl alcohol in the blood [51]. Therefore, "after consumption" is understood as the content of ethyl alcohol in the blood from 0.20 to 0.50‰ or from 0.10 to 0.25 mg/L, while the "intoxication state" is considered to be the content of ethyl alcohol in the blood of above 0.50‰ or 0.25 mg/L [52].

On the basis of the control protocols, developed by inspectors of the National Labor Inspectorate, we were able to determine the exact time at which the accident occurred. On the basis of the time of the accident and the content of ethyl alcohol in the blood, we were able to estimate the amount of consumed alcohol. This alcohol was calculated per 500 mL of 4.5% beer consumed by an employee (one "large beer" is equivalent to the consumption of 175mL of 10.0% wine or 50 mL of 40.0% vodka). Figure 2 presents the Polish standard drink.

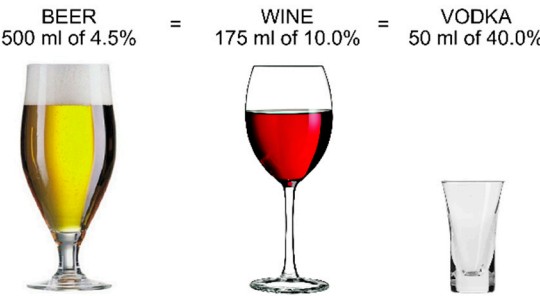

**Figure 2.** The Polish standard drink (own elaboration).

In order to determine:

- the maximum blood alcohol content (in ‰);
- the time after finishing drinking and reaching the maximum value (min);
- the time after finishing drinking and reaching the value of 0.00 ‰, i.e., total sobriety (min);

We used the ethyl alcohol content calculator [53]. It is widely known that the metabolism of alcohol involves individual differences [54]. The alcohol metabolism efficiency of different individuals will vary greatly [55]. Therefore, the reference unit was adopted for analysis. It was assumed that the reference unit is a man with a statistical height of a Pole equal to 1.80 m, a statistical body weight of 83 kg [11], and at the age that corresponds to the average age obtained in the analyses—40 years. This man has a normal body shape, normal alcohol consumption, and standard food consumption.

Figure 3 presents a graph of the change in the content of ethyl alcohol in the blood with regards to the dose of consumed alcohol (from 500 mL of 4.5% beer—1 beer, to 5500 mL of 4.5% beer—11 beers) and the time that elapsed since the end of drinking. For example, after consuming three beers (1500 mL of

4.5%) or three glasses of wine (525 mL of 10.0%) or three glasses of vodka (150 mL of 40.0%), the highest content of ethyl alcohol in blood (1.18‰)/blood alcohol concentration (BAC = 0.12 g/100 mL) content is 90 min after the end of drinking. The human body needs 480 min for content of ethyl alcohol in blood 0.0‰ (BAC = 0.00 g/10 0mL), which means the person is sober. In addition, Table 1 presents the characteristic data resulting from Figure 3. The Table 1 indicates the time when the content of ethyl alcohol in the blood is the highest and the time since the end of drinking indicating 0‰.

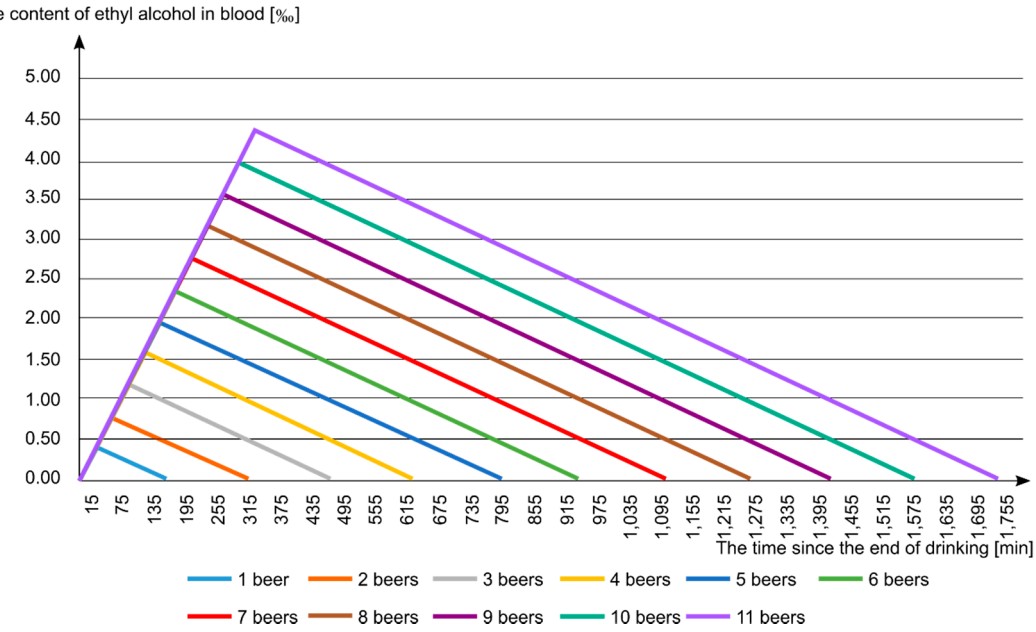

**Figure 3.** The content of ethyl alcohol in the blood with regards to the dose of alcohol consumed (from 1 beer to 11 beers) and the time since the end of drinking (own elaboration).

**Table 1.** Collective summary of the content of ethyl alcohol in the blood with regards to the dose of consumed alcohol (from 1 beer to 11 beers) and the time since the end of drinking (own elaboration).

| Alcohol Dose | | | Content of Ethyl Alcohol in Blood | Blood Alcohol Concentration (BAC) | Time Since the End of Drinking |
|---|---|---|---|---|---|
| Number of 4.5% Beers/(mL) | Number of 10.0% Glasses of Wine/(mL) | Number of 40.0% Glasses of Vodka/(mL) | (‰) | (g/100 mL) | (min) |
| 1/500 | 1/175 | 1/50 | 0.39 0.00 | 0.04 0.00 | 30 160 |
| 2/1000 | 2/350 | 2/100 | 0.78 0.00 | 0.08 0.00 | 60 320 |
| 3/1500 | 3/525 | 3/150 | 1.18 0.00 | 0.12 0.00 | 90 480 |
| 4/2000 | 4/700 | 4/200 | 1.56 0.00 | 0.16 0.00 | 120 640 |
| 5/2500 | 5/875 | 5/250 | 1.95 0.00 | 0.21 0.00 | 150 800 |
| 6/3000 | 6/1050 | 6/300 | 2.34 0.00 | 0.25 0.00 | 180 960 |
| 7/3500 | 7/1225 | 7/350 | 2.73 0.00 | 0.29 0.00 | 210 1120 |
| 8/4000 | 8/1400 | 8/400 | 3.12 0.00 | 0.34 0.00 | 240 1280 |
| 9/4500 | 9/1575 | 9/450 | 3.51 0.00 | 0.38 0.00 | 270 1440 |
| 10/5000 | 10/1750 | 10/500 | 3.90 0.00 | 0.42 0.00 | 300 1600 |
| 11/5500 | 11/1925 | 11/550 | 4.290.00 | 0.460.00 | 3301760 |

The following scenarios were adopted in the conducted analyses:

- Scenario 1: the injured person consumed alcohol on the day preceding the accident, while assuming that drinking finished at 21:00;
- Scenario 2: the injured person consumed alcohol during the work break preceding the accident. Three specific cases were assumed: (a) alcohol was consumed before starting work, i.e., 7:00; (b) alcohol was consumed during the breakfast break at 10:00; (c) alcohol was consumed during the lunch break at 13:00.

The main point of the scenarios above was to determine the amount consumed alcohol by the workers and the time of drinking. To determine the above hours, we used the Expert (Delphi) method. This method, by means of which obtained results were based upon, relied upon the opinions and assessments of competent experts [56]. The adopted assumptions resulted directly from the answers received from construction managers, work managers, and supervision inspectors about experience with working with construction workers. The experts were asked to, on the basis of their own experience and knowledge, present occurring situations or scenarios for workers under the influence of alcohol on their supervised construction sites. On the basis of the analysis of the information received from the expert, two different types of scenarios emerged: first—consumed alcohol on the day preceding the accident, and second—consumed alcohol during the work break preceding the accident.

### 3.2. Surveys

Surveys constituted the second source of data. The survey data were collected from January 2016 to December 2018 during the on-going research project called "Model of the assessment of risk of the occurrence of building catastrophes, accidents and dangerous events at workplaces with the use of scaffolding" ("ORKWIZ"). The surveys and questionnaires were prepared by a team from the Lublin University of Technology, under the leadership of K. Czarnocki [57]. From 1500 people working on the examined 120 construction sites (during the testing of 120 scaffoldings), 573 employees took part in the study (573 surveys were carried out among people working at the construction site of the examined scaffoldings).

Participation in the survey was voluntary and anonymous. Respondents had the right to refuse to participate without giving a reason. All the procedures performed in studies involving human participants were in accordance with the ethical standards and with the 1964 Helsinki Declaration and its later amendments [58]. According to the current guidelines of the Ethical Review Board at the Centre of Postgraduate Medical Education, Warsaw, Poland [59], an anonymous questionnaire-based cross-sectional study does not require separate consent.

On the basis of the information received from construction management, the researchers estimated that a total of 1500 people worked on the 120 examined construction scaffoldings. In 120 initial surveys, in the part concerning the scaffolding assembly team, the researchers asked employees about their age, seniority, and experience in scaffolding assembly. In contrast, in 573 personal surveys—covering 38.2% of the employees working on the scaffoldings in question—people were asked about drugs they used (alcohol, cigarettes, and other intoxicants).

The data collection method included questionnaires. The data obtained in this study were direct responses from individuals. As part of the research, the researchers developed standardized protocols for data collection. Furthermore, all study personnel were trained to conduct the research. It is well known that training of study personnel allows for the minimization of inter-observer variability [60]. The questions concerned private or sensitive topics, such as consumption of alcohol. Thus, self-reporting data may have been affected by an external bias caused by social desirability. The bias in this case can be referred to as social desirability bias [61]. Moreover, at the stage of validation, the data obtained from the questionnaire were analyzed using the methods of descriptive statistics in order to verify the variability of responses to individual questions. What is important is that the analysis of the collected

data was performed after all the surveys had been completed. This approach was intended to reduce the interviewer's bias [62].

Respondents in the personal survey were first asked if they had ever consumed alcohol, and then whether they had consumed alcohol in the last 12 months. In the research, a person who did not drink alcohol was defined as a person who had never consumed alcohol, or a person who had consumed alcohol, but not in the last 12 months. If the respondents gave both positive answers, they were asked further questions about their frequency of drinking. An answer to the asked question was the amount of consumed "standard" alcoholic beverages, i.e., 4.5% beers with a capacity of 500 mL—"one large beer" during a day. The obtained data were subjected to detailed analysis, taking into account such parameters as age, marital status, and also the place of residence of the employees participating in the survey.

In the case of the age criterion, the following categories were applied: 18–19 years, 20–29 years, 30–39 years, 40–49 years, 50–59 years and >60 years. Marital status was classified as single, married, divorced, or widower. Permanent residence was classified as follows: village, a small town with up to 100,000 residents, and a city of over 100,000 residents.

The data were analyzed with Statistica v.13.3 (StatSoft Polska Sp z o.o.). Normality of distributions of continuous variables [63] was assessed by the Shapiro–Wilk test [64]. The distribution of categorical variables was shown by frequencies and proportions along with 95% confidence intervals [65]. Associations between personal characteristics (age, marital status, permanent residence) with drinking alcohol were conducted using the logistic regression analyses [66]. The socio-demographic characteristics were considered as independent variables [67]. In univariate logistic regression analyses [68], we considered all variables separately. Statistical inference was based on the criterion $p < 0.05$ [69].

## 4. Research Results

### 4.1. Analysis of Occupational Accidents

The control reports concerning 219 people injured in occupational accidents at workplaces that use building scaffoldings were analyzed with regards to the causes of the accidents. An interesting factor for the authors of this study was the human cause—consumption of alcohol, narcotic drugs, or psychotropic substances [70]. The cause related to alcohol consumption occurred in 38 injured people, which was 17.4% of all people injured in accidents on scaffolding. This means that every sixth occupational accident was caused by an abnormal sobriety of an employee.

In 22 control protocols, labor inspectors determined the exact value of the alcohol content in the blood of the injured person. In the remaining 16 cases, the protocol only provided information about the cause of the accident—alcohol consumption.

A detailed analysis of the alcohol content in a victim's blood showed that the lowest amount of ethyl alcohol in the blood was equal to 0.20‰ (state after consumption indicates alcohol consumption with typical symptoms of diffuse attention), while the highest identified value was equal to 4.16‰ (intoxication with typical symptoms such as balance disorder, speech disorder, drowsiness, decreased behavior, and movement control, and also impairment of hand–eye coordination). The average alcohol content value in a victim's blood was equal to 1.20 ± 1.10‰. Therefore, the state after consumption of alcohol was determined in the case of four people, while the remaining 18 people were in the state of intoxication.

The age structure of people for whom the accident was caused by alcohol is shown in Figure 4. The most numerous group was employees aged 40–49 (34.2% of all people with alcohol in their blood) and 30–39 (31.58%). It was in these age ranges (30–49) that the highest levels of ethyl alcohol were found in the blood. The average age of victims was 40 ± 8 years. It can be stated that the percentage of alcohol-consuming workers increases with age.

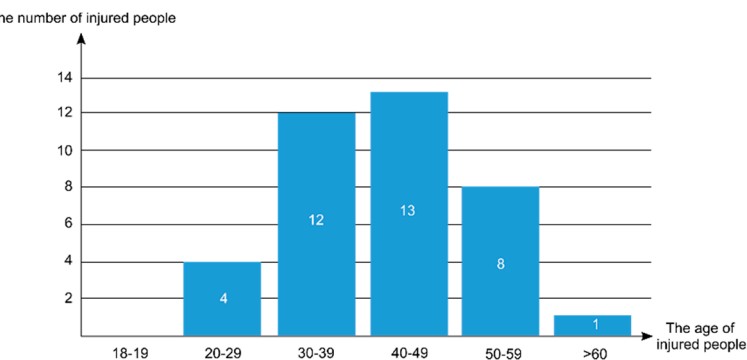

**Figure 4.** Age structure of injured people who were under the influence of alcohol based on control protocols (own elaboration).

The effect of the accident was also analyzed. Accidents were divided into fatal, severe, and light, and are shown in Figure 5. The age structure is as follows:

- Eight people had a fatal accident, which was 21% of the analyzed accidents. The alcohol content in five people for whom the blood alcohol content was known ranged from 0.53–4.16‰ (mean value of 1.97 ± 0.94‰). This means that all injured people were intoxicated (alcohol content greater than 0.50‰).
- Seventeen people had a severe accident, which represented 45% of all the analyzed cases. The alcohol content in 10 people for whom their blood alcohol content was known ranged from 0.22–3.40‰ (mean value of 1.47 ± 0.81‰). Two of them were in a state after the consumption of alcohol (alcohol content of less than 0.50‰), and the remaining eight people were drunk.
- Thirteen people suffered a light accident, which represented 34% of all the analyzed cases. The alcohol content in seven people for whom the blood alcohol value was known ranged from 0.44–3.00‰ (mean value 1.37 ± 0.86‰). Two of them were in a state after the consumption of alcohol (alcohol content of less than 0.50‰), and the remaining five people were drunk.

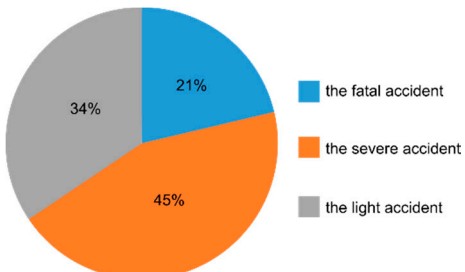

**Figure 5.** The result of an accident of people under the influence of alcohol on the basis of control protocols (own elaboration]).

On the basis of the control protocols, we were also able to determine the exact time at which the accident occurred. The analysis was carried out for 22 accidents for which the time of the accident and the content of alcohol in the victim's blood were known. Unreal situations were rejected for each accident, e.g., consumption of more than 5000 mL of 4.5% beer—10 beers (life-threatening situation). The most probable scenario was selected.

Table 2 presents the values of alcohol consumption with regards to the content of ethyl alcohol in the blood (‰), the time of the accident, the adopted scenario, and the estimated number of consumed beers. If the analysis resulted in an impossible situation (e.g., the time since the end of alcohol consumption and the blood alcohol content being impossible), the table indicates such a case with the abbreviation "*im*". The table highlights the assumed numbers of consumed alcoholic beverages in relation to 500 mL of 4.5% beer ("large beers").

**Table 2.** Estimated numbers of consumed alcoholic beverages ("large beers") (own elaboration).

| No. | Severity | Content of Alcohol in Blood | Scenario 1 | | Scenario 2 (a) | | (b) | | (c) | |
|---|---|---|---|---|---|---|---|---|---|---|
| | | | Time | Number of Beers | Time | Number of Beers | Time | Number of Beers | Time | Number of Beers |
| (-) | (-) | (‰) | (min) | (pcs) | (min) | (pcs) | (min) | (pcs) | (min) | (pcs) |
| 1 | L | 0.70 | 900 | 7 | 270 | 3 | 90 | 2 | - | - |
| 2 | S | 2.13 | 1080 | 10 | 480 | 7 | 300 | 6 | 120 | im |
| 3 | L | 0.53 | 720 | 5 | 120 | 2 | - | - | - | - |
| 4 | S | 1.23 | 680 | 7 | 80 | im | - | - | - | - |
| 5 | S | 1.85 | 1020 | 10 | 420 | 6 | 240 | 5 | 60 | im |
| 6 | S | 1.30 | 690 | 7 | 90 | 3 | - | - | - | - |
| 7 | L | 0.22 | 1070 | 7 | 470 | 3 | 290 | 2 | 110 | 1 |
| 8 | F | 3.00 | 810 | >10 | 210 | im | 30 | im | - | - |
| 9 | F | 4.16 | 1075 | >10 | 475 | 11 | 295 | im | - | - |
| 10 | S | 1.33 | 900 | 8 | 300 | 4 | 120 | 3 | - | - |
| 11 | S | 1.11 | 860 | 8 | 260 | 4 | 80 | 3 | - | - |
| 12 | F | 1.70 | 1070 | 10 | 470 | 6 | 290 | 5 | 110 | im |
| 13 | L | 1.00 | 840 | 7 | 240 | 3 | 60 | 2 | - | - |
| 14 | L | 0.46 | 1020 | 7 | 420 | 3 | 240 | 2 | 60 | 1 |
| 15 | L | 0.44 | 1020 | 7 | 420 | 3 | 240 | 2 | 60 | 1 |
| 16 | S | 1.30 | 885 | 8 | 285 | 4 | 105 | 3 | - | - |
| 17 | L | 0.20 | 780 | 5 | 180 | 2 | - | - | - | - |
| 18 | F | 2.74 | 990 | >10 | 390 | 8 | 210 | 7 | 30 | im |
| 19 | F | 3.40 | 1110 | >10 | 510 | 10 | 330 | 9 | 150 | im |
| 20 | S | 2.67 | 840 | 10 | 240 | 7 | 60 | im | - | - |
| 21 | S | 2.00 | 1110 | >10 | 510 | 7 | 330 | 6 | 150 | im |
| 22 | S | 0.63 | 610 | 5 | 10 | im | - | - | - | - |

Information about the severity of the accident is present in Table 2 next to the number (No.) in the second column:

- "L" means light accident;
- "S" means severe accident;
- "F" means fatal accident.

For the authors of the study, nine cases (cases 2, 5, 8, 9, 12, 18, 19, 20, 21) aroused great concern and doubt—the obtained blood alcohol content was high enough (i.e., higher than 1.70‰ and amounting to 2.13‰, 1.85‰, 3.00‰, 4.16‰, 1.70‰, 2.74‰, 3.40‰, 2.67‰, 2.00‰, respectively) that allowing an employee to work in a life-threatening condition would be extremely irresponsible. Such an employee would have, and the authors had no doubts about this, come to work in a drunken state, which should have been immediately noticed by the construction supervisor (site manager, foreman). Further analysis of the causes of these accidents indicated the second cause of the accident, i.e., the lack of direct supervision over the operation that was performed by the injured party.

Other cases and the most probable scenarios indicate that

- injured parties were likely to drink alcohol during the work break immediately before the accident: scenario (2a)—cases: 6, 17 (1000–1500 mL of 4.5% beer: 2–3 beers), scenario (b)—cases: 1, 10, 11, 13, 16 (500–1500 mL of 4.5% beer: 1—3 beers), scenario (c)—cases: 7, 14, 15 (500 mL of 4.5% beer: 1 beer);
- injured parties most likely consumed alcohol the day before (scenario 1)—cases: 3, 4, 22 (2500–3500 mL of 4.5% beer: 5—7 beers).

### 4.2. Analysis of Survey Data

From 1500 people working on the examined 120 construction sites, 573 employees took part in the study. This was 38.3% of the people employed in the examined construction enterprises. The researchers tested 120 façade scaffoldings. The façade scaffoldings were divided into four groups with regards to the surface area: 30–300 m$^2$ (55 scaffolding), 300–600 m$^2$ (28 scaffolding), 600–900 m$^2$ (25 scaffolding), and 900—1500 m$^2$ (12 scaffolding). Figures 6–9 show examples of tested scaffoldings with regards to the surface area.

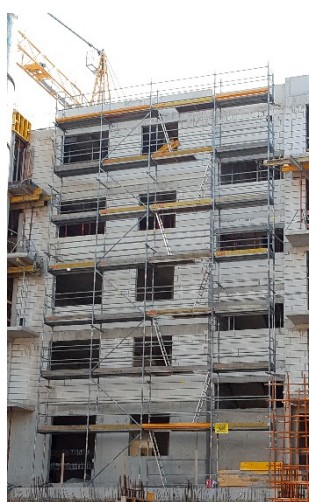

**Figure 6.** Example of façade scaffold with an area of 30–300 m$^2$ (authors' archive).

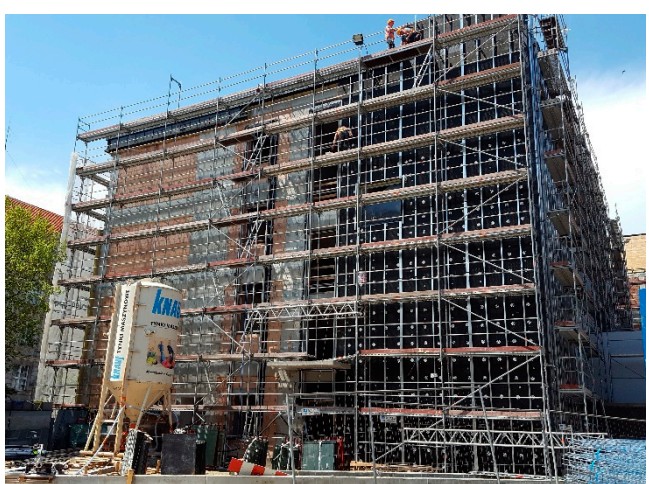

**Figure 7.** Example of façade scaffold with an area of 300–600 m$^2$ (authors' archive).

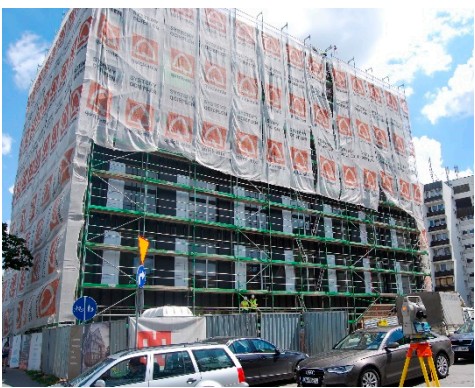

**Figure 8.** Example of façade scaffold with an area of 600–900 m$^2$ (authors' archive).

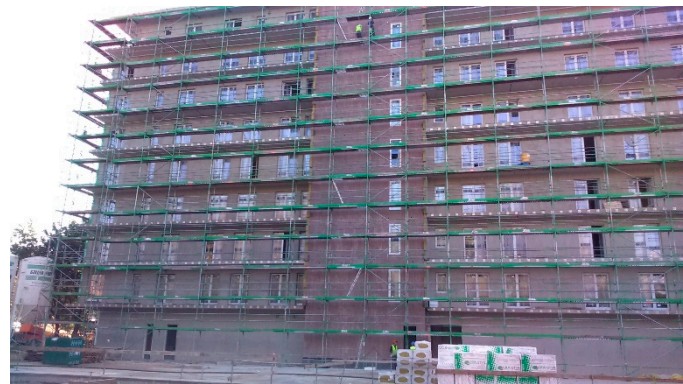

**Figure 9.** Example of façade scaffold with an area of 900–1500 m$^2$ (authors' archive).

Table 3 presents the obtained socio-demographic information on the respondents.

**Table 3.** Socio-demographic information (own elaboration).

| Age of Employee | | | |
|---|---|---|---|
| **Age Range** | **Number of People** | **Percentage** | ***p*** |
| 18–19 | 13 | 2.3% | |
| 20–29 | 178 | 31.1% | |
| 30–39 | 209 | 36.5% | <0.01 |
| 40–49 | 102 | 17.8% | |
| 50–59 | 53 | 9.2% | |
| >60 | 18 | 3.1% | |
| **Marital Status** | | | |
| Marital Status | Number of People | Percentage | *p* |
| Single | 251 | 43.8% | |
| Married | 304 | 53.1% | <0.01 |
| Divorced | 11 | 1.9% | |
| Widower | 7 | 1.2% | |
| **Permanent Residence** | | | |
| Place | Number of People | Percentage | *p* |
| A city of over 100,000 residents | 179 | 31.2% | |
| A small town with up to 100,000 residents | 204 | 35.6% | <0.01 |
| Village | 190 | 33.2% | |

Figure 10 shows the age structure of the respondents. The employment structure is as follows:

- The largest group of respondents were employees aged 30–39, and they represented 36.5% of all the respondents. They were mainly employees with average experience who had worked for at least 10 years in their profession.
- The second group was made up of young employees aged 20–29 (31.1%), with a much shorter seniority, ranging from several days to a maximum of 10 years.
- Other respondents were experienced employees with more than 15 years of work experience at the age of 40–49 (17.8%) and 50–59 (9.2%), several employees over the age of 60 who were often at retirement age (3.1%), and adolescent employees (2.3%).

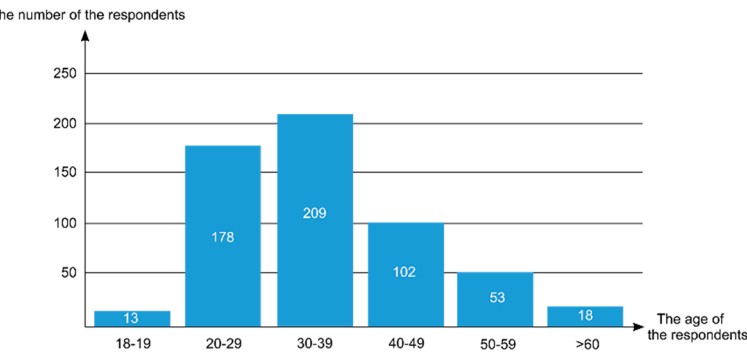

**Figure 10.** The age structure of the respondents (own elaboration).

The majority of the respondents were married (53.1% of all respondents), or bachelors (43.8%), 35.6% of whom indicated small towns as their place of permanent residence (up to 100,000 inhabitants), 33.2% of whom lived in villages, and 31.2% of whom lived in cities (over 100,000 inhabitants) ($p < 0.01$).

Figure 11 shows the obtained survey data concerning the consumption of alcohol by 573 construction employees who worked on the 120 assessed construction scaffoldings.

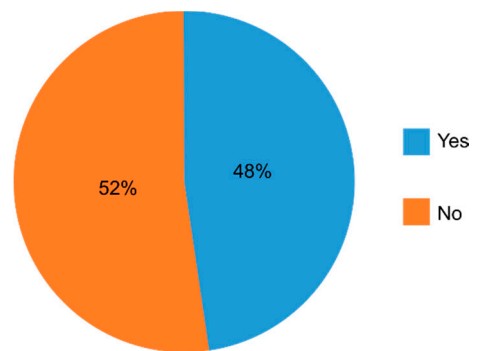

**Figure 11.** The consumption of alcohol by construction workers (own elaboration).

A total of 274 people (i.e., 47.8% of surveyed employees) declared that they consumed alcohol during the day, while the remaining 299 people (52.2%) declared that they had never consumed alcohol, or had consumed alcohol, but not during the last 12 months. During the research, no employee declared alcohol abuse, alcoholism, or alcohol consumption at work.

One of the questions that was asked in the survey concerned the number of consumed "standard" alcoholic beverages, i.e., 4.5% beers with a capacity of 500 mL—one "large beer". The obtained answers showed that the largest number of "large beers" consumed during a day (500 mL) was equal to 10 and was declared by nine people. The most common answer was one beer, and such information was given by 144 respondents, i.e., 25.1% of all the respondents. The remaining people declared the quantities that are shown in Figure 12. The average value for the studied population was 1000 mL ± 1000 mL of 4.5% beer (2 ± 2 beers).

The analysis of the place of residence showed that most often people declared living in the countryside—99 people, and in small towns (up to 100,000 inhabitants)—95 people. A total of 80 people declared that they permanently lived in cities (over 100,000 inhabitants).

The age of people who consumed alcohol, as well as the age of abstainers, was also analyzed. Table 4 presents the number of people with regards to their age.

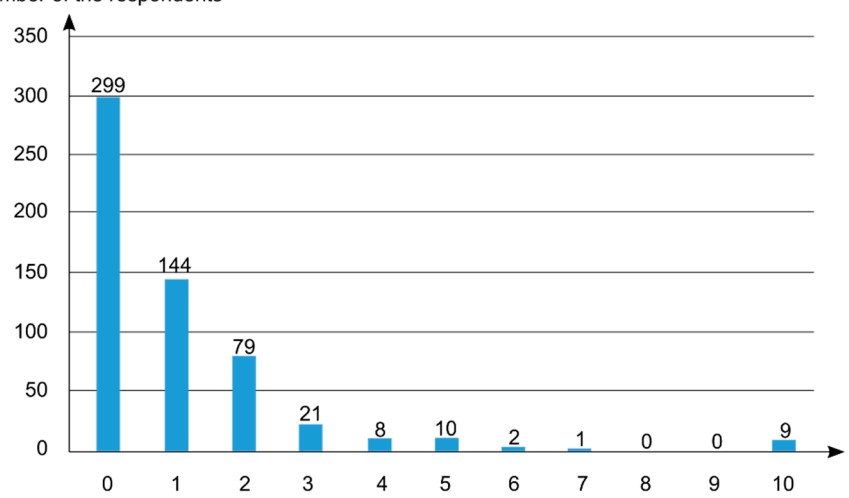

**Figure 12.** Number of consumed alcoholic beverages ("large beers") declared by construction workers during a day (own elaboration).

**Table 4.** The number of people consuming alcohol ("Yes"), and the number of abstainers ("No"), with regards to their age (own elaboration).

| Age Range | Number of People | Answer | Number of People | Percentage |
|-----------|------------------|--------|------------------|------------|
| 18–19 | 13 | Yes | 3 | 23.1% |
|  |  | No | 10 | 76.9% |
| 20–29 | 178 | Yes | 85 | 47.8% |
|  |  | No | 93 | 52.2% |
| 30–39 | 209 | Yes | 100 | 47.8% |
|  |  | No | 109 | 52.2% |
| 40–49 | 102 | Yes | 47 | 46.1% |
|  |  | No | 55 | 53.9% |
| 50–59 | 53 | Yes | 29 | 54.7% |
|  |  | No | 24 | 45.3% |
| >60 | 18 | Yes | 10 | 55.6% |
|  |  | No | 8 | 44.4% |

The following conclusions can be drawn as a result of the obtained data (Table 4):

- The most numerous group who consumed alcohol during the day were employees aged 30–39 (100 people) and 20–29 (85 people);
- The highest percentage of people consuming alcohol occurred among employees aged over 60 (55.6%) and 50–59 (54.7%);
- The number of people consuming alcohol decreased with age (decreasing tendency) and the ratio of people consuming alcohol to the abstainers changed (the number of people consuming alcohol increased);
- Among younger employees (18–19, 20–29, 30–39), the percentage of people consuming alcohol was smaller than the percentage of abstainers.

During the tests, photographic documentation of the construction site was also carried out. It evidences that alcohol was consumed at the examined sites, e.g., Figure 13 shows an abandoned beer can at a litter site at the examined construction site, and Figure 14 shows a beer can in the assembly yard at the examined construction site.

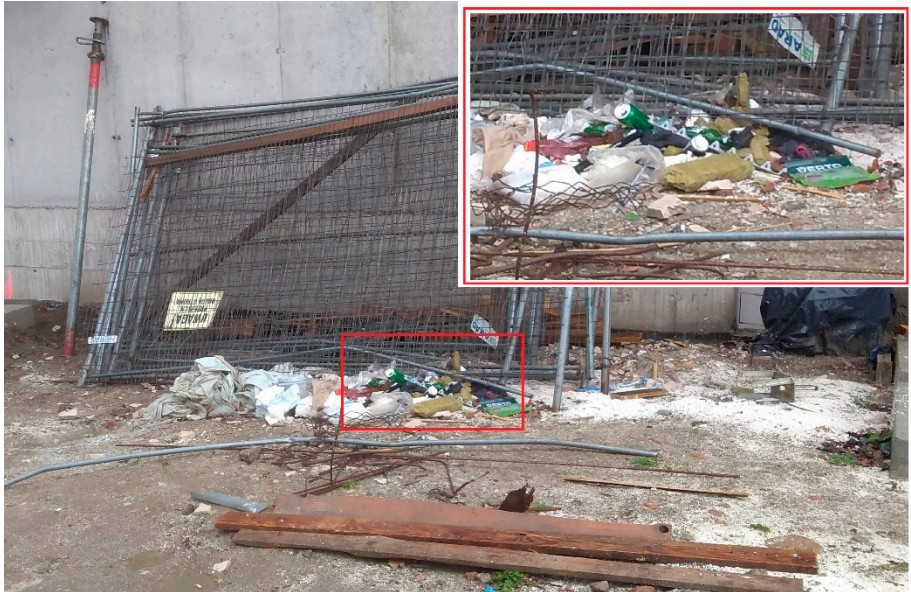

**Figure 13.** An abandoned beer can at a litter site at the examined construction site (authors' archive).

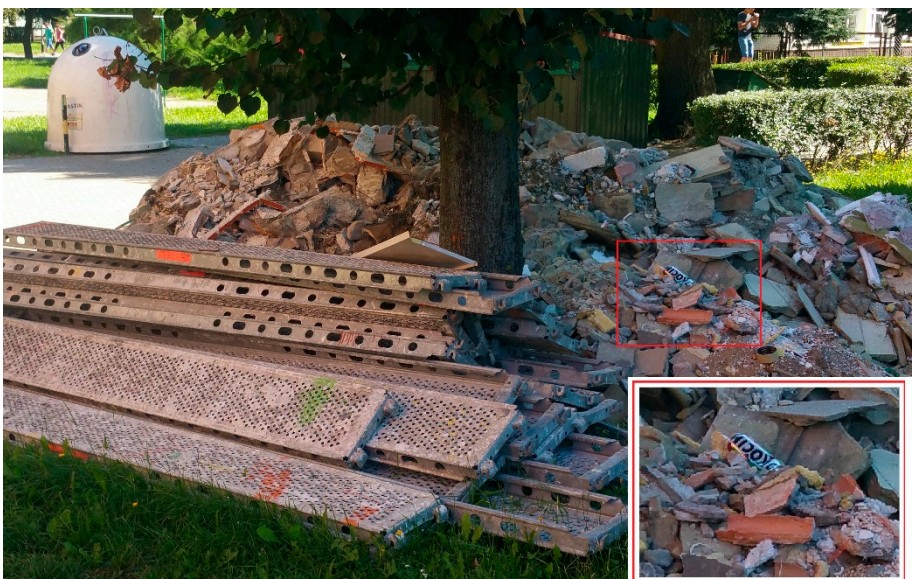

**Figure 14.** Beer can in the assembly yard at the examined construction site (authors' archive).

### 4.3. Summary of Results

Analysis of 219 accident control protocols concerning occupational accidents in the construction industry involving construction scaffolding, which took place in Poland, allowed the following conclusions to be drawn:

- Alcohol is one of the causes that leads to occupational accidents. In the analyzed accidents on scaffoldings, the cause related to alcohol consumption occurred in 17.4% of all the victims. This means that every sixth occupational accident was caused by an abnormal sobriety of employees
- The largest group of people consuming alcohol were employees aged 40–49 and 30–39. The average age of victims was 40 ± 8 years. The proportion of drinkers increased with age.
- Analysis of the content of alcohol in blood with regards to the effect of an accident indicated that as the amount of alcohol in the blood rises, the severity of the accident increases. A high alcohol content contributes to the death of a construction worker.

An analysis of 573 construction employees working on the assessed 120 construction scaffoldings allowed the following conclusions to be formulated:

- 47.8% of surveyed employees declared that they drink alcohol during the day;
- the most common number of alcohol beverages consumed outside working hours was one "large beer" (500 mL of 4.5% beer);
- analysis of the place of residence showed that people most often declared living in the countryside and in small towns (up to 100,000 inhabitants);
- the proportion of people consuming alcohol increased with age (an increasing trend of consuming alcohol with age), and the proportion of people consuming alcohol to abstainers changed—the number of people consuming alcohol increased.

The place where an employee is working is constantly and dynamically changing [71]. With such dynamic environments, construction sites contain a significant quantity of unidentified or not well-assessed hazards that expose construction workers to additional safety risks during required operations [72]. Working on scaffolding carries a much higher risk of hazards than other types of construction works [73]. The main reason is the place of work itself—work at a height. Working from a height continues to be one of the major causes of fatality within the construction industry [74].

## 5. Discussion

The study identified the main problems associated with the abuse and consumption of alcohol at work among employees in the construction industry, with particular emphasis on jobs related to work on scaffolding. When the authors of the article examined the susceptibility of individual age groups to alcohol consumption, the pattern of alcohol consumption by construction workers in Poland emerged. It was found that every second person consumes alcohol during the day, and the number of people consuming alcohol increases with age.

Five limitations of our study should be mentioned. Firstly, it is worth noting that the accident documentation did not always contain all the necessary information about the content of ethyl alcohol in the blood of the injured person. Because the accident documentation (inspection reports) was prepared by inspectors of the National Labor Inspectorate, the authors were unable to interfere or supplement this archival data. It happened that such information was not provided/confirmed by the police officer or doctor admitting the injured person to the hospital ward. In the analyzed data, the cause related to alcohol consumption occurred in 38 injured people, which was 17.4% of all people injured in accidents, while the content of ethyl alcohol in the blood of the injured person was included only in 22 control protocols. Secondly, the content of ethyl alcohol in the blood was determined, assuming the reference unit. It should be remembered that the metabolism of alcohol involves individual differences [54]. The alcohol metabolism efficiency of different individuals will vary greatly [55]. Thirdly, the second source of the data analyzed was obtained on the basis of surveys. The proportion that took part in the survey was 38.2% of the target population. The data obtained in this study were direct responses from individuals. It may be assumed that the number of consumed "standard" alcoholic beverages might have been even higher than what the data revealed. No data were available regarding the comparisons of respondents and non-respondents. Since there was no officially recorded data, results obtained from the data collection method could be biased. Findings might have been subject to selection bias, although the data were collected with the greatest care. Fourth, since the second source of the data was collected via a survey and actively answered by working individuals, it did not contain data related to fatal accidents. Fifth, during the conducted research on the 120 examined construction scaffoldings, the authors did not collect any photographic documentation of some workers drinking an alcoholic beverage, for example, a can of beer. This was hard to do because the workers knew that the research team was conducting research and thus their behavior was close to "perfect". Moreover, while examining the scaffolding and conducting the survey, the authors did not register the situation

that some workers were drinking alcoholic beverages. This may have been due to the fact that the construction workers consumed alcohol in hard-to-reach places on the construction site (in hiding).

The observations carried out also revealed that there was an additional, serious problem that had not yet been analyzed, which is the wide availability of alcoholic beverages. Unfortunately, a very unfavorable factor that affects the ease of drinking alcohol immediately before work or during breaks from work is the large, easy, and widespread availability of alcohol. It is worth noting that both in Poland and in other European countries, alcohol can be bought in different volumes and in every discount store or hypermarket. The assortment is very wide and available in various volumes, starting with the smallest bottles of 50 mL of 40.0% vodka (the shape of the bottle enables it to fit in a pocket), through to bottles of 200 mL of 40.0% vodka (measuring 7.8 × 17.0 cm with a flat shape that allows it to be stored, e.g., in a jacket pocket) and larger bottles (500 mL, 700 mL, and more). According to the authors, the availability of small-volume bottles should be limited, as this could help reduce the number of people who drink alcohol at work.

In addition, according to the authors, in order to improve the safety of employees in a workplace and to eliminate the problem of intoxicated construction workers, employers (including the construction manager and work managers) should be allowed to carry out sobriety checks of employees. Currently, an employer has the right to carry out an inspection only if two conditions are met: first—an employee has agreed to conduct such an examination (the examination is voluntary), and second—an employer has reasonable suspicion that an employee is under the influence of alcohol [75]. Therefore, at present, an employer cannot carry out such an examination if an employee does not agree to it. An employer has the right to request such an examination to be conducted by an authorized body, e.g., the police service. An employer is also not able to conduct routine sobriety checks on all or randomly selected employees.

## 6. Conclusions

The construction industry is recognized as one of the most dangerous industries. Much effort has been devoted to improving safety and to reduce hazards in the workplace, but less attention has been paid to the human factor, i.e., employees in the workplace. It should be remembered that the most important thing when considering safety at work is people. This study confirmed that alcohol consumption negatively affects the human body; reduces the ability to properly and safely, i.e., fault-free and accident-free, perform standard daily activities (such as driving, moving) and professional activities (e.g., work in an office, work on a construction site, work on scaffolding); and can also lead to death at a workstation.

Data for the analysis were obtained from two sources. The archival post-accident documentation, which was the first source of data from 2008–2017, allowed for the determination of the most probable scenario of alcohol consumption by employees during work. The advantage of these studies was the 10-year period of data collection, which allowed for the establishment of a certain trend in the accident situations. Unfortunately, the data were prepared by various inspectors of the National Labor Inspectorate and contained varying degrees of detail (from 38 post-accident documentation, only 16 protocols stated that the "injured person consumed alcohol" or was "under the influence of alcohol"). Therefore, when planning this type of research, it should be remembered that we may be dealing with incomplete data when using archival data. On the other hand, surveys (the second source of data) required the researchers to properly plan and conduct surveys. When preparing the questionnaire, it is important to remember that the questions should be logical, understandable to the respondents, and not be suggestive of answers. It is also particularly important to properly train the study personnel. The advantage of these studies was the testing of nearly 40% of people working at workplaces with scaffoldings.

This study confirmed that excessive and disproportionate alcohol consumption can be the cause of an accident, and consequently death (depending on the type of accident or physical ailment), at workplaces with scaffolding. Of 219 accident reports, 17.4% indicated alcohol as a contributing factor. Analysis of accident documentations shows that cases where alcohol was indicated as a contributing

factor in an accident, and that the alcohol was consumed during the workday. Furthermore, the analysis of the blood alcohol content with regards to the effect of an accident indicates that with an increase in the amount of alcohol in the blood, the severity of the accident also increases. Comparative analysis of the results from the post-accident documentation and survey showed that the number of people consuming alcohol decreased with age, but the number of alcohol-related accidents did not decrease. Moreover, the percentage of people consuming alcohol slightly changed with age.

The research was conducted in five research areas, i.e., in five provinces (voivodeships) of Poland (Dolnośląskie, Lubelskie, Łódzkie, Mazowieckie, Wielkopolskie). The research of post-accident documentation showed that most of the accidents related to alcohol took place in central Poland, i.e., in the Łódzkie Voivodeship (12 accidents, i.e., that every four accidents were related to alcohol). On the other hand, the surveys showed that most of the people consuming alcohol worked on construction sites in the central and eastern part of Poland, i.e., in the Mazowieckie Voivodeship (59% of working people) and Lubelskie Voivodeship (57% of working people). Moreover, according to the answers provided by the respondents, the largest number of people who did not drink while working (i.e., abstinents) was in the Wielkopolskie Voivodeship (71%).

The use of alcohol is important a topic of occupational safety in the construction industry. This research is relevant to the development of interventions for reduction of occupational accidents and is of interest to public health. The results obtained on the basis of the conducted research are able to constitute a justification for the directions of preventive actions carried out in order to reduce the number of occupational accidents in the construction industry caused by alcohol. This will significantly contribute to an increase of the level of occupational safety in the construction industry.

**Author Contributions:** Conceptualization, M.S. (Marek Sawicki); data curation, M.S. (Marek Sawicki) and M.S. (Mariusz Szóstak); formal analysis, M.S. (Marek Sawicki) and M.S. (Mariusz Szóstak); investigation, M.S. (Marek Sawicki) and M.S. (Mariusz Szóstak); methodology, M.S. (Marek Sawicki) and M.S. (Mariusz Szóstak); visualization, M.S. (Mariusz Szóstak); writing—original draft preparation, M.S. (Marek Sawicki) and M.S. (Mariusz Szóstak); writing—review and editing, M.S. (Marek Sawicki) and M.S. (Mariusz Szóstak). All authors have read and agreed to the published version of the manuscript.

**Funding:** The article is the result of the implementation by the authors of the research project no. 244388 "Model of the assessment of risk of the occurrence of building catastrophes, accidents and dangerous events at workplaces with the use of scaffolding", financed by NCBiR within the framework of the Programme for Applied Research on the basis of contract no. PBS3/A2/19/2015.

**Conflicts of Interest:** The authors declare no conflict of interest.

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
