# Peer review of "Impact of Alcohol on Occupational Health and Safety in the Construction Industry at Workplaces with Scaffoldings"

_applsci, doi:10.3390/app10196690_

Round 1
Reviewer 1 Report
Article Title
The title is correct and seems to clearly identify the purpose of the document. But it is better to identify the expression "Safety" with "Health and Safety".
Abstract
Paragraph lines 14 to 17 - Regarding the way of expressing, please do not repeat concepts already indicated previously. The expression "data" has been repeated three times on two lines. The information that is transmitted regarding the data is not clear. It is necessary to indicate that the sources were different "... from two different sources." Subsequently, the type of information from both sources must be clarified. "The first one was post-accident documentation on occupational accidents. The second one was surveys collected during the research project." Please briefly explain what kind of interviews, about what kind of people and construction sites. Do not reference a research project in quotation marks. You can say: "This research project is based on a risk assessment model ...". I recommend that you use the methodology section to include the reference of this research project.
Line 18. - We all know that alcohol consumption does not cause death. I think this concept should be properly qualified. It can be affirmed that excessive and disproportionate consumption can be the cause of an accident, and consequently death (depending on the type of accident or physical ailment). Please modify the phrase on line 18 and, in addition, specify in what type of job.
Lines 19 and 20 - The following expression is said: "... that the alcohol was consumed during the workday." How is this information obtained: interview of the research project, interview of the doctors, autopsy ...? not clear. Please explain this concept very briefly.
Lines 21, 22 and 23 - "It should be ... ... public health". They are generic sentences that have nothing to do with the research or the purpose of this article. They are sentences that can be used at the beginning as an introduction to the Abstract. For example: "The value, care and customs of workers are essential in terms of occupational health and safety. This implies the reduction of accidents and increased interest for public health. However, the abuse of alcohol consumption is generally considered ... etc. " This is an example. Please consider your best option to rewrite the text.
Lines 23, 24 and 25 - "The results ... ... by alcohol". The mention made of the results is too generic. There are even doubts about its application: "... conducted research may be a justification ..." May be ...? If you are confident in your research and the results it provides, please be more categorical in your statements.
The abstract is understood but not clear in focus. You should consider rewriting the abstract. Remember that the abstract must briefly include introduction, objective, methodology, results and conclusion. Consider the indications that are proposed to you.
Keywords
The keywords might be correct but not all are correct. "Work Safety" is incorrect. Better "Health and Safety" and "workplace". The term "construction scaffoldings" is ambiguous. Are you referring to scaffolding construction or scaffolding workplaces? It is better to use "Construction industry" and "scaffolding".
- Introduction
Line 30 - Regarding your bibliographic reference [2] (Appleton, 2018), I transcribe the first sentence: "With alcohol being a leading cause of death worldwide it is a considerable public health concern ..." It is possible that the translation may generate some confusion. Alcohol may be one of the causes but not the main cause. Saying "... a leading cause ..." is not the same as saying "... the leading cause ...". I recommend these two links below. Please consider rewriting the sentence: "Alcohol is the leading cause of death worldwide."
https://www.who.int/news-room/fact-sheets/detail/the-top-10-causes-of-death
https://www.who.int/substance_abuse/publications/global_alcohol_report/en/
Line 30,31,32 - "According ... ... with it [3,4]." Regarding their bibliographic references [3 and 4] (The National Program for Prevention of Alcohol Problems for the years 2011-2015 (in Polish). 2010) and (Anderson, 2006), they refer to Poland and Europe, respectively. However, reference is made to the World Health Organization. It should be differentiated in the text when reference is made to a country, a continent, or the world. Please consider modifying this text and its references (even updating them).
Line 36,37 - "According ... ... is significant." In the introduction, and less at the beginning of it, it is better not to make references to the research itself. So this sentence must be cited. This type of problem is identified by the World Health Organization, so it is certain that there are significant references, not only for Poland, but for different countries. I recommend that you rewrite this statement and use appropriate references.
Line 38 - "... physical effort," the sentence must be cited.
Line 39 - "... weather conditions." the sentence must be cited.
Line 40 - "... for relaxation." the sentence must be cited.
Line 43,44 – “…consequences.” the sentence must be cited.
Lines 44,45 – “…economic consequences,” this sentence must be cited.
Line 49 – “6 litres per person in 2002 to 10 litres in 2018.” Is this consumption per day, per week, per month or per year ...?
Lines 51,52,53 - "In addition, according ... ... 12% of the population." If you are comparing results it is better to keep the same criteria: alcoholics - 2% - xxx thousand people; and alcohol abusers - 12% - yyy thousand people.
Although his research focuses on data taken in Poland, it is important to add in the introduction section what is happening in the countries around Poland, including in the countries of Europe, in order to offer a more adequate perspective of this enormous problem.
The introductory texts deal with the problem of alcohol consumption without particularizing the purpose of its investigation. In the introductory section, no reference was made to the purpose of the research: the impact of alcohol consumption in jobs with scaffolding. It is recommended that text cited and referring to the construction sector, workplaces and jobs that depend on a scaffold be added. Of course, all referring to the risks that are generated by alcohol co nsumption.
Line 54 – “…impact on human health…” this sentence must be cited.
Paragraph 54-58 - There are two objectives: one main (overarching goal) and another one (the second goal). It is recommended that you indicate in advance that there are multiple objectives or two objectives. then the first, and the second aim; or the overarching, and the particular aim.
The introduction is very poor and does not focus on the purpose of the research. For the article to be published, it is required to rewrite according to the indications given above. The reasons for your research purpose should be added to the objectives that are raised. Finally, what is the purpose of the research and the intended use of it.
- Literature review
Paragraph 65-67 and paragraph 68-77 - It is exposed on the risk of accident due to alcohol consumption and its consequences due to excess consumption. They are particular issues. They should be after exposing the statistics of the different countries for alcohol consumption.
Line 152-153 - Please do not use the term "demonstrated". No formula has been proven. You could use the expression: "Based on the Literature review .." or whatever you think best. This sentence must be cited.
Paragraph 163-167 - is the same paragraph as lines 54 to 58. Please do not repeat the same statements.
Line 168 - The following expression is said: "In the study concerning alcohol consumption patterns ..." It is not clear whether it refers to the purpose of this investigation or is a reference to another investigation. Please clarify the sentence.
This section is well founded. But the structure is not adequate and it does not keep pace. Situations and results are mixed. A good structure could be:
- General alcohol consumption (worldwide, other countries and Poland);
- the consumption of alcohol in the workplace in general (worldwide, other countries and Poland);
- alcohol use in construction jobs (worldwide, other countries and Poland);
- Risks associated with alcohol consumption in construction jobs (worldwide, other countries and Poland).
This is a proposal. You must propose an organized structure from the general topic to the particular topic of your research. Finally, you can make the appropriate observations, justified by the texts, to expose the objectives of the research.
- Methodology of research
In the methodology section, the two procedures are poorly explained. It is surprising that photographs of the investigation are not shown: some alcohol consumption measuring glasses, some of the scaffolds on which an inspection and survey has been carried out, where the scaffolding works are located (in the city, in towns ... ), height of scaffolding, type of construction, etc.
3.1. Accident documentation
Line 196 – “…ethyl alcohol in the blood.” This sentence must be cited.
Line 200 - Who determines the Control Protocols; the Government, the University, your research ...? this sentence must be cited.
Line 208-209 - Sentence: “Widely known that the metabolism of alcohol involves individual differences.” this sentence must be cited.
Line 209-201 - Sentence: “The alcohol metabolism efficiency of different individuals will vary greatly.” this sentence must be cited.
Figure 1 - You should better explain this graph. Explain the results of the graph for a specific data; for example for the consumption of three beers.
Table 1 - The line for the consumption of 11 beers is missing.
Line 224 - Why were two scenarios adopted? You must explain the procedure and justify
Line 232 - What is the Expert method? You should briefly explain this methodology and justify its use. This method must be cited.
The data collection procedure from the first source of information should be explained
3.2. Surveys
Line 239-240 - The phrase: "During the testing of 120 scaffoldings, 573 surveys were carried out among people working at the construction site of the examined scaffoldings." it should go to the end of the paragraph.
Line 243-247 - Both references must be cited.
Line 253-254 - The phrase "From 1,500 people working on the examined 120 construction sites, 573 employees took part in the study." It should be at the end of paragraph 236-241. You should rewrite paragraph 236-241.
Line 261-262 - It is not necessary to repeat the consumption equivalence data, in parentheses. It has already been specified above.
Paragraph 269-275 - You must cite all the sources you are referencing (I think there are 7 references in this paragraph). Please be clearer in your explanations in the methodology section. This paragraph should be expanded to justify the use of each of the sources. You will need to include an application example.
- Research results
4.1. Analysis of occupational accidents
Line 332 - Table 2 indicates very interesting information. It should take two lines and explain the meaning of their corresponding values to each column. This paragraph should be incorporated before table 2.
4.2. Analysis of survey data
Line 350 - It is indicated that 573 surveys have been carried out. Has a survey been carried out on the 219 people injured in a work-related accident or on any of them?
Evaluator's comment - The results offered are based on a sample of 120 inspected works and 573 people. It could be very interesting to elaborate some results based on the type of construction: small, medium or large works. Or even, based on the type of company: small, medium or large company.
Keep in mind that you have a large number of results and may get different interpretations based on observation points. It is very important to be able to establish preventive actions to improve the problem of alcohol consumption. But I think your results are too generic and do not show the reality of the problem.
Line 391 - In the description of table 4 it is better that you add the following: "Table 4. The number of people consuming alcohol (Yes), and the number of abstainers (No), with regards to their age [own elaboration].” Or provide a clarification or explanation.
Line 403 - Surely many photographs have been taken in each of the inspections carried out for data collection. However, the photographs shown only show the existence of beer cans. However, the photographs show the accumulation of debris as an element of risk. It is more interesting to show the types of buildings and scaffolds that have been inspected. It would have been better to provide a photograph with some workers drinking a can of beer. The faces in the photo can be blurred. I consider that the photographs are not representative. I should add some photos. If you consider the size of the photos to be inconvenient, you could reduce it. 8 photos could be placed on the page at a legible size.
4.3. Summary of results
This section shows a brief summary of the results obtained in the investigation. These are very interesting facts. However, on the one hand, in order to use this information, it is necessary to establish the type of accidents and the injuries or injuries they have caused (consider the need to incorporate this information). And, on the other hand, if it can be compared with other investigations in order to determine the degree of danger that working in a job with scaffolding has with respect to other types of jobs (this part could be referenced with the appropriate bibliography).
- Discussion
Lines 443-447 - The sentences described are not the result of your investigation. Each of the sentences should be cited.
Paragraph 448-454 - Each of the sentences must be cited, as it is not the result of your research. If the citation [39] refers to the entire paragraph, please, it is better that at the beginning of the paragraph describe who is the author of this research: "An investigation carried out by Liu et al [39], determined the following parameters ... "
It is not understood why the discussion section makes references to other research. If citations are provided, it is to serve as a discussion towards the results of your research. You should incorporate data from your research that discusses with data from other researches. Please, if you do not have results you will have to move from line 437 to line 454 to the Literature Review section. Also, warn you that the quote [39] refers to China. You should consider expanding this information with quotes from closer countries in order to establish a proper basis for discussion.
Line 455 - The expression "Therefore ..." determines a continuity in the content of the section. You cannot establish a continuity in the discussions when the previous two paragraphs are quotes and are not the result of your research.
Lines 466-468 - The sentence "It should be ... ... differences." must be cited. and the sentence "The alcohol ... ... vary greatly." must be cited.
Lines 470-472 - The following clarification is made: "Since there is no officially recorded data, results obtained from the data collection method could be biased. Findings might have been subject to selection bias, although the data was collected with the greatest care. " I recommend you delete this clarification. To do this, you will have to give an explanation in the methodology section about how the data was collected and the data selected in the surveys. Please note that this statement could lead to the article not being published. For this, there are different types of psychosocial standards that guarantee the veracity of the answers. That is why it is important that you explain how the survey was developed.
Lines 480-481 - "It is worth noting here that in Poland, unlike other European countries..." I recommend that you do not make such an inappropriate reference. It is better that you make references to certain countries (quoting) or to several countries in general. For example: "It is worth noting that both in Poland and in other European countries ..."
- Conclusions
The conclusions that are proposed do not refer to the results of the investigation nor do they propose possible solutions.
From a general approach, a particular conclusion can be proposed regarding your research.
The statement (line 505): "... should be allowed to carry out sobriety checks of employees." What does the law say about it? This is more of a discussion than a conclusion.
Line 508 - there is an appointment; the [40]. No bibliographic references are made in the conclusions section. They are conclusions from the results of your research.
It is very difficult to change customs. It is best to pose educational conditions at different age levels. Of course, start reporting the consequences of drinking alcohol in the workplace, in elementary schools.
The conclusions they contribute to this research are not relevant and should be rewritten. It has to be based on your own research, its results and the benefits it can bring to society.
References
In references, it is important that you add the DOI information (if any) for each item. For example, the first reference:
[1] Miller, T.R .; Zaloshnja, E .; Spicer, R.S. Effectiveness and benefit-cost of peer-based workplace substance abuse prevention coupled with random testing. Accid. Anal. Prev. 2007, 39, 565–573. https://doi.org/10.1016/j.aap.2006.10.001
The percentage of current references is good, but remember that the new references that you will have to add must be current as well.
Author Response
Dear Reviewer,
Thank you very much for your review and your critical comment, which allow to do our article better. We apologize that the previous version of our paper did not meet your expectations. We hope that in the current version of the paper, we have taken into account all your critical remarks. We also hope, that the current version meets your expectations.
Below we present the point-by-point response to received reviews and the answers to the most important comments and suggestions for Authors.
Comment: Article Title The title is correct and seems to clearly identify the purpose of the document. But it is better to identify the expression "Safety" with "Health and Safety".
Answer: Thank you very much for your comment. We added the expression “Health and Safety”.
Comment: Abstract Paragraph lines 14 to 17 - Regarding the way of expressing, please do not repeat concepts already indicated previously. The expression "data" has been repeated three times on two lines. The information that is transmitted regarding the data is not clear. It is necessary to indicate that the sources were different "... from two different sources." Subsequently, the type of information from both sources must be clarified. "The first one was post-accident documentation on occupational accidents. The second one was surveys collected during the research project." Please briefly explain what kind of interviews, about what kind of people and construction sites. Do not reference a research project in quotation marks. You can say: "This research project is based on a risk assessment model ...". I recommend that you use the methodology section to include the reference of this research project.
Answer: Thank you very much for your comment. We changed this section as follow:
Data for the analysis was obtained from two different sources. The first one was post-accident documentation on occupational accidents (analysis of a group of 219 people injured in occupational accidents that involved construction scaffolding in 2008-2017). The second one was surveys collected during the research project. This research project is based on a risk assessment model of the occurrence of building catastrophes, accidents and dangerous events at workplaces with the use of scaffolding (analysis of 573 surveys of people working on construction sites involving 120 scaffoldings).
Comment: Line 18. - We all know that alcohol consumption does not cause death. I think this concept should be properly qualified. It can be affirmed that excessive and disproportionate consumption can be the cause of an accident, and consequently death (depending on the type of accident or physical ailment). Please modify the phrase on line 18 and, in addition, specify in what type of job.
Answer: Thank you very much for your comment. We changed this section as follow:
This study confirmed that excessive and disproportionate alcohol consumption can be the cause of an accident, and consequently death (depending on the type of accident or physical ailment) at workplaces with scaffolding.
Comment: Lines 19 and 20 - The following expression is said: "... that the alcohol was consumed during the workday." How is this information obtained: interview of the research project, interview of the doctors, autopsy ...? not clear. Please explain this concept very briefly.
Answer: Thank you very much for your comment. We changed this section as follow:
Of 219 accident reports, 17.4% indicated alcohol as a contributing factor. Analysis of accident documentations shows that n cases where alcohol was indicated as a contributing factor in an accident, that the alcohol was consumed during the workday.
Comment: Lines 21, 22 and 23 - "It should be ... ... public health". They are generic sentences that have nothing to do with the research or the purpose of this article. They are sentences that can be used at the beginning as an introduction to the Abstract. For example: "The value, care and customs of workers are essential in terms of occupational health and safety. This implies the reduction of accidents and increased interest for public health. However, the abuse of alcohol consumption is generally considered ... etc. " This is an example. Please consider your best option to rewrite the text.
Answer: Thank you very much for your comment. We added your suggestions and changed the order of the sentences.
Comment: Lines 23, 24 and 25 - "The results ... ... by alcohol". The mention made of the results is too generic. There are even doubts about its application: "... conducted research may be a justification ..." May be ...? If you are confident in your research and the results it provides, please be more categorical in your statements.
Answer: Thank you very much for your comment. We changed this section as follow:
Furthermore, the analysis of the blood alcohol content with regards to the effect of an accident indicates that with an increase in the amount of alcohol in the blood, the severity of the accident also increases.
The results obtained on the basis of the conducted research able to constitute a justification for the directions of preventive actions carried out in order to reduce the number of occupational accidents in the construction industry caused by alcohol.
Comment The abstract is understood but not clear in focus. You should consider rewriting the abstract. Remember that the abstract must briefly include introduction, objective, methodology, results and conclusion. Consider the indications that are proposed to you.
Answer: Thank you very much for your proposed suggestions. The whole abstract was changed as follow:
Abstract: The value, care and customs of workers are essential in terms of occupational health and safety. This implies the reduction of accidents and increased interest for public health. However, the abuse of alcohol is widely regarded as a serious threat to the lives, health and safety of employees. The aim of the research was to identify the main problems that are associated with alcohol abuse and consumption at work among employees in the construction industry, with particular emphasis on workstations where work is carried out on construction scaffoldings.
Data for the analysis was obtained from two different sources. The first one was post-accident documentation on occupational accidents (analysis of a group of 219 people injured in occupational accidents that involved construction scaffolding in 2008-2017). The second one was surveys collected during the research project. This research project is based on a risk assessment model of the occurrence of building catastrophes, accidents and dangerous events at workplaces with the use of scaffolding (analysis of 573 surveys of people working on construction sites involving 120 scaffoldings).
This study confirmed that excessive and disproportionate alcohol consumption can be the cause of an accident, and consequently death (depending on the type of accident or physical ailment) at workplaces with scaffolding. Of 219 accident reports, 17.4% indicated alcohol as a contributing factor. Analysis of accident documentations shows that n cases where alcohol was indicated as a contributing factor in an accident, that the alcohol was consumed during the workday. Furthermore, the analysis of the blood alcohol content with regards to the effect of an accident indicates that with an increase in the amount of alcohol in the blood, the severity of the accident also increases.
The results obtained on the basis of the conducted research able to constitute a justification for the directions of preventive actions carried out in order to reduce the number of occupational accidents in the construction industry caused by alcohol.
Comment: Keywords The keywords might be correct but not all are correct. "Work Safety" is incorrect. Better "Health and Safety" and "workplace". The term "construction scaffoldings" is ambiguous. Are you referring to scaffolding construction or scaffolding workplaces? It is better to use "Construction industry" and "scaffolding".
Answer: Thank you very much for your comment. We changed keywords as follow:
health and safety; workplace; occupational accidents; construction industry; scaffoldings; drugs; alcohol.
Comment: Introduction Line 30 - Regarding your bibliographic reference [2] (Appleton, 2018), I transcribe the first sentence: "With alcohol being a leading cause of death worldwide it is a considerable public health concern ..." It is possible that the translation may generate some confusion. Alcohol may be one of the causes but not the main cause. Saying "... a leading cause ..." is not the same as saying "... the leading cause ...". I recommend these two links below. Please consider rewriting the sentence: "Alcohol is the leading cause of death worldwide."
https://www.who.int/news-room/fact-sheets/detail/the-top-10-causes-of-death
https://www.who.int/substance_abuse/publications/global_alcohol_report/en/
Answer: Thank you very much for your comment. We changed this sentence and added new references:
[3] World Health Organization. Global status report on alcohol and health 2018. ISBN 978-92-4 156563-9, https://www.who.int/substance_abuse/publications/global_alcohol_report/en/
Comment: Line 30,31,32 - "According ... ... with it [3,4]." Regarding their bibliographic references [3 and 4] (The National Program for Prevention of Alcohol Problems for the years 2011-2015 (in Polish). 2010) and (Anderson, 2006), they refer to Poland and Europe, respectively. However, reference is made to the World Health Organization. It should be differentiated in the text when reference is made to a country, a continent, or the world. Please consider modifying this text and its references (even updating them).
Answer: Thank you very much for your comment. We modified and updated references as follow:
[3] World Health Organization. Global status report on alcohol and health 2018. ISBN 978-92-4 156563-9, https://www.who.int/substance_abuse/publications/global_alcohol_report/en/
Comment: Line 36,37 - "According ... ... is significant." In the introduction, and less at the beginning of it, it is better not to make references to the research itself. So this sentence must be cited. This type of problem is identified by the World Health Organization, so it is certain that there are significant references, not only for Poland, but for different countries. I recommend that you rewrite this statement and use appropriate references.
Answer: Thank you very much for your comment. We modified and updated references as follow:
[7]Roche, A.; Chapman, J.; Duraisingam, V.; Phillips, B.; Finnane, J.; Pidd, K. Construction workers’ alcohol use, knolwedge, perceptions of risk and workplace norms. Drug and Alcohol Review 2020, https://doi.org/10.1111/dar.13075
[8] Strickland, J.; Wagan, S.; Dale, A.; Evanoff, A. Prevalence and perception of risky health behaviors among construction workers. J. Occup. Environ. Med. 2017, 59(7), 673-678, DOI: 10.1097/JOM.0000000000001051
Comment: Line 38 - "... physical effort," the sentence must be cited.
Answer: Thank you very much for your comment. We modified and updated references as follow:
[9] Hoła, B.; Szóstak, M. An occupational profile of people injured in accidents at work in the polish construcion industry. Procedia Eng. 2017, 208, 43-51, https://doi.org/10.1016/j.proeng.2017.11.019
Comment: Line 39 - "... weather conditions." the sentence must be cited.
Answer: Thank you very much for your comment. We modified and updated references as follow:
[10] Hoła, B.; Nowobilski, T.; Szer, I.; Szer, J. Identification of factors affecting the accident rate in the construction industry. Procedia Eng. 2017, 208, 35-42, https://doi.org/10.1016/j.proeng.2017.11.018
Comment: Line 40 - "... for relaxation." the sentence must be cited.
Answer: Thank you very much for your comment. We modified and updated references as follow:
[11] Lim, S.; Chi, S.; Lee, J.; Lee, H-J.; Choi, H. Analyzing psychological consitions of field-workers in the construction industry. Int. J. Occup. Environ. Health 2017, 23(4), 261-281, https://doi.org/10.1080/10773525.2018.1474419
Comment: Line 43,44 – “…consequences.” the sentence must be cited.
Answer: Thank you very much for your comment. We modified and updated references as follow:
[14] Ken, P.; Vinita, D.; Roche, A.; Allan, T. Young construction workers: substance use, mental health, and workplance psychosocial factors. Adv. Dual. Diagn. 2017, 10(4), 155-165, doi:http://dx.doi.org/10.1108/ADD-08-2017-0013
Comment: Lines 44,45 – “…economic consequences,” this sentence must be cited.
Answer: Thank you very much for your comment. We modified and updated references as follow:
[16] Tiwary, G.; Gangopadhyay, P.; Biswas, S.; Nayak, K.; Chatterjee, M.; Chakraborty, D.; Mukherjee, S. Socio-economic status of workers of building construction industry. Indian J. Occup.Environ. Med. 2012, 16(2), 66-71. doi:http://dx.doi.org/10.4103/0019-5278.107072
Comment: Line 49 – “6 litres per person in 2002 to 10 litres in 2018.” Is this consumption per day, per week, per month or per year ...?
Answer: Thank you very much for your comment. We added more detail as follow:
According to this data, the level of alcohol consumption increased from approx. 6 litres per person per year in 2002 to 10 litres in 2018.
Comment: Lines 51,52,53 - "In addition, according ... ... 12% of the population." If you are comparing results it is better to keep the same criteria: alcoholics - 2% - xxx thousand people; and alcohol abusers - 12% - yyy thousand people.
Answer: Thank you very much for your comment. We added more detail as follow:
In addition, according to data in Poland, alcoholics constitute about 2% of the population, i.e. 600-800 thousand people, and alcohol abusers constitute about 12% of the population, i.e. 3.6-4.8 million people.
Comment: Although his research focuses on data taken in Poland, it is important to add in the introduction section what is happening in the countries around Poland, including in the countries of Europe, in order to offer a more adequate perspective of this enormous problem.
Answer: Thank you very much for your comment. We added more detail about countries around Poland as follow:
Statistical data published by the Statistical Office of the European Union (Eurostat) are the basis for the statement that the serious problem of alcohol consumption exists [24] Only 16.4% of male in 2014 declared that “never or not in the last 12 months” consumption alcohol. This value is the frequency of alcohol consumption for 27 countries of the European Union (EU27). And so, "every day" and "every week" alcohol consumption declared by 14.7% and 35.5% of men in the EU27. The highest daily frequency of alcohol consumption was in Portugal (38.6%), and the weekly frequency in the United Kingdom (51.6%).
[24] Statistical Office of the European Union, Available online: https://ec.europa.eu/eurostat/web/health/data/database (accessed on 10 August 2020).
Comment: The introductory texts deal with the problem of alcohol consumption without particularizing the purpose of its investigation. In the introductory section, no reference was made to the purpose of the research: the impact of alcohol consumption in jobs with scaffolding. It is recommended that text cited and referring to the construction sector, workplaces and jobs that depend on a scaffold be added. Of course, all referring to the risks that are generated by alcohol consumption.
Answer: Thank you very much for your comment. We added more detail about the construction sector, workplaces and jobs that depend on a scaffold as follow:
The phenomenon of alcohol abuse and consumption at work is a common problem in many sectors of the economy, especially in the construction [19]. Many construction workers have alcohol‐related problems [20,21]. This problem also applies to workers at workplaces with scaffoldings [22]. Alcohol increases the risk of accident situations. Moreover, the most common events that cause accidents to workers on scaffolding are falls from height caused by among others reduced concentration [23].
[19] Biggs, H.; Williamson, A. Reducing the risk of alcohol and other drugs in construction: an Autralian assessment. In New Developments in Structural Engineering & Construction, Research Publishing Services, Singapore; 2013; pp. 1399-1404. doi: 10.3850/978-981-07-5354-2_CS-14-255.
[20] Banwell, C.; Dance, P.; Quinn, C.; Davies, R.; Hall, D. Alcohol, other drug use, and gambling among Australian Capital Territory (ACT) workser in the building and related industries. Drugs (Abingdon Engl) 2006, 13(2), 167-178, https://doi.org/10.1080/09687630600577550.
[21] Coggon, D.; Harris, E.; Brown, T.; Rice, S.; Palmer, K. Occupation and mortality related to alcohol, drugs and sexual habits. Occup. Med. 2010, 60(5), 348-353, https://doi.org/10.1093/occmed/kqq040.
[22] Rebelo, M.; Silveira, F.; Czarnocka, E.; Czarnocki, K. Construction safety on scaffolding; Building Information Modeling (BIM)_ and Safety Management – a systematic review. U.Porto J. Eng. 2019, 2, 46-60, https://doi.org/10.24840/2183-6493_005.002_0006.
[24] Hoła, B.; Szóstak, M. Modeling of the accidentality phenomenon in the construction industry. Appl. Sci. 2019, 9, 1878, https://doi.org/10.3390/app9091878.
Comment: Line 54 – “…impact on human health…” this sentence must be cited.
Answer: Thank you very much for your comment. We modified and updated references as follow:
[15] Wang, S.-C.; Chen, Y.-C.; Chen, S.-J.; Lee, C-H.; Cheng, C.-M. Alcohol addiction, gut microbiota, and alcoholism treatment: a review. Int. J. Mol. Sci. 2020, 21(17), 6413, https://doi.org/10.3390/ijms21176413.
Comment: Paragraph 54-58 - There are two objectives: one main (overarching goal) and another one (the second goal). It is recommended that you indicate in advance that there are multiple objectives or two objectives. then the first, and the second aim; or the overarching, and the particular aim.
Answer: Thank you very much for your comment. We added new information about aim of this study as follow:
There were two aims in this study. First: the overarching goal of the research was to identify the main problems associated with the consumption of alcohol at work among employees in the construction industry, with particular emphasis on jobs related to work on scaffolding. Second: the particular goal of the study was to determine patterns of alcohol consumption among construction workers in Poland.
Comment: The introduction is very poor and does not focus on the purpose of the research. For the article to be published, it is required to rewrite according to the indications given above. The reasons for your research purpose should be added to the objectives that are raised. Finally, what is the purpose of the research and the intended use of it.
Answer: Thank you very much for your comment. We hope that in the current version of the paper, we have taken into account all your critical remarks. Besides the resubmission of the manuscript we are also enclosing the paper in the track changes which may be handy for the reviewers to spot the areas of our improvements in this section.
Comment: Literature review Paragraph 65-67 and paragraph 68-77 - It is exposed on the risk of accident due to alcohol consumption and its consequences due to excess consumption. They are particular issues. They should be after exposing the statistics of the different countries for alcohol consumption.
Answer: We do not agree with Your comment. According to the Authors it is not necessary to present the statistics of the different countries for alcohol consumption in relation to this issues, because each paragraphs and sentences have appropriate citations.
Comment: Line 152-153 - Please do not use the term "demonstrated". No formula has been proven. You could use the expression: "Based on the Literature review .." or whatever you think best. This sentence must be cited.
Answer: Thank you very much for your comment. We changed this text and added references as follow:
[29] Peacock, A.; Leung, J.; Larney, S.; Colledge, S.; Hickman, M.; Rehm, J.; Giovino, G.; West, R.; Hall, W.; Griffths, P.; Ali, R.; Gowing, L.; Marsden, J.; Ferrari, A.; Grebely, J.; Farrell, M.; Degenhardt, L. Global statistics on alcohol, tobacco and illicit drug use: 2017 status report. Addiction 2018, 113(10), 1905-1926, doi: 10.1111/add.14234.
Comment: Paragraph 163-167 - is the same paragraph as lines 54 to 58. Please do not repeat the same statements.
Answer: Thank you very much for your comment. We removed this text.
Comment: Line 168 - The following expression is said: "In the study concerning alcohol consumption patterns ..." It is not clear whether it refers to the purpose of this investigation or is a reference to another investigation. Please clarify the sentence.
Answer: Thank you very much for your comment. We changed this text - it refers to the purpose of this investigation.
Comment: This section is well founded. But the structure is not adequate and it does not keep pace. Situations and results are mixed. A good structure could be:
- General alcohol consumption (worldwide, other countries and Poland);
- the consumption of alcohol in the workplace in general (worldwide, other countries and Poland);
- alcohol use in construction jobs (worldwide, other countries and Poland);
- Risks associated with alcohol consumption in construction jobs (worldwide, other countries and Poland).
This is a proposal. You must propose an organized structure from the general topic to the particular topic of your research. Finally, you can make the appropriate observations, justified by the texts, to expose the objectives of the research.
Answer: Thank you very much for your comment. We changed the structure, added new paragraphs and the order of the text. We hope that in the current version of the paper, we have taken into account all your critical remarks. Besides the resubmission of the manuscript we are also enclosing the paper in the track changes which may be handy for the reviewers to spot the areas of our improvements in this section.
Comment: Methodology of research In the methodology section, the two procedures are poorly explained. It is surprising that photographs of the investigation are not shown: some alcohol consumption measuring glasses, some of the scaffolds on which an inspection and survey has been carried out, where the scaffolding works are located (in the city, in towns ... ), height of scaffolding, type of construction, etc.
Answer: Thank you very much for your comment. We changed this section accordance with the following remarks and answers.
Comment: Accident documentation Line 196 – “…ethyl alcohol in the blood.” This sentence must be cited.
Answer: Thank you very much for your comment. We modified and updated references as follow:
[50] Koob, G.; Arends, M.; Moal, M. Drugs, addiction, and the brain, 2014, Chapter 6 – Alcohol, 173-219, Academic Press, https://doi.org/10.1016/B978-0-12-386937-1.00006-4.
Comment: Line 200 - Who determines the Control Protocols; the Government, the University, your research ...? this sentence must be cited.
Answer: Thank you very much for your comment. We added more detail as follow:
Based on the Control Protocols, developed by inspectors of the National Labour Inspectorate, it was possible to determine the exact time at which the accident occurred.
Comment: Line 208-209 - Sentence: “Widely known that the metabolism of alcohol involves individual differences.” this sentence must be cited.
Answer: Thank you very much for your comment. We modified and updated references as follow:
[54] Caderbaum, A. Alcohol metabolism. Clin. Liver. Dis. 2012, 16(4), 667-685, https://doi.org/10.1016/j.cld.2012.08.002.
Comment: Line 209-201 - Sentence: “The alcohol metabolism efficiency of different individuals will vary greatly.” this sentence must be cited.
Answer: Thank you very much for your comment. We modified and updated references as follow:
[55] Edenberg, H. The Genetics of alcohol metabolism: role of alcohol dehydrogenase and aldehyde dehydrogenase variants. Alcohol Res. Health, 2007, 30(1), 5-13.
Comment: Figure 1 - You should better explain this graph. Explain the results of the graph for a specific data; for example for the consumption of three beers.
Answer: Thank you very much for your comment. We added more detail as follow:
For example, after consuming 3 beers (1,500 ml of 4.5%) or 3 glasses of wine (525 ml of 10.0%) or 3 glasses of vodka (150 ml of 40.0%) the highest content of ethyl alcohol in blood (1.18‰)/Blood Alcohol Concentration (BAC=0.12g/100ml) content is 90 minutes after the end of drinking. The human body needs 480 minutes for content of ethyl alcohol in blood 0,0‰ (BAC=0.00 g/100ml), which means the person is sober.
Comment: Table 1 - The line for the consumption of 11 beers is missing.
Answer: Thank you very much for your comment. We added missing data.
Comment: Line 224 - Why were two scenarios adopted? You must explain the procedure and justify
Answer: Thank you very much for your comment. We added more detail as follow:
The experts were asked to, based on their own experience and knowledge, presented occurring situations or scenarios for workers under the influence of alcohol on supervised their construction sites. Based on the analysis of the information received from the expert, two different types of scenarios have emerged: first - consumed alcohol on the day preceding the accident and second - consumed alcohol during the work break preceding the accident.
Comment: Line 232 - What is the Expert method? You should briefly explain this methodology and justify its use. This method must be cited.
Answer: Thank you very much for your comment. We added more detail as follow:
The main point of the above scenarios was determine the amount consumed alcohol by the workers and time of drinking. To determine the above hours the Expert (Delphi) method was used. This method by means of which obtained results are based on the opinions and assessments of competent experts [56].
[56] Iriste, S.; Katane, I. Expertise as a research method in education, Rural Environ. Educ. Person. 2018, 11, 74-80, https://doi.org/10.22616/REEP.2018.008.
Comment: The data collection procedure from the first source of information should be explained
Answer: The data collection procedure from the first source was explained at the beginning of this section. We believe that such a description is sufficient.
Based on the Control Protocol, i.e. based on a description of the circumstances and causes of the accident, it was possible to obtain information about the health status of an injured person during and after the accident. The report includes information, confirmed by a police officer or doctor admitting the injured person to the hospital ward, concerning blood alcohol content (in units: promil or mg/l), the consumed amount of alcoholic beverages, and also the statement: "state after consumption", i.e. indicating alcohol consumption or "intoxication state".
Comment: Surveys Line 239-240 - The phrase: "During the testing of 120 scaffoldings, 573 surveys were carried out among people working at the construction site of the examined scaffoldings." it should go to the end of the paragraph.
Answer: Thank you very much for your comment. We rewrote this paragraph as follow:
From 1,500 people working on the examined 120 construction sites (during the testing 120 scaffoldings), 573 employees took part in the study (573 surveys were carried out among people working at the construction site of the examined scaffoldings).
Comment: Line 243-247 - Both references must be cited.
Answer: We added the both references:
[58] World Mecidal Association, WMA Declaration of Helsinki – ethical pricniples for medical research involving human subjects, 1964 as amended, Available online: https://www.wma.net/policies-post/wma-declaration-of-helsinki-ethical-principles-for-medical-research-involving-human-subjects/ (accessed on 10 August 2020).
[59] The Supreme Medical Court, The Polish Code of Medical Ethics, 2003 (in Polish). Available online: https://nil.org.pl/dokumenty/kodeks-etyki-lekarskiej(accessed on 10 August 2020).
Comment: Line 253-254 - The phrase "From 1,500 people working on the examined 120 construction sites, 573 employees took part in the study." It should be at the end of paragraph 236-241. You should rewrite paragraph 236-241.
Answer: Thank you very much for your comment. We rewrote this paragraph as follow:
From 1,500 people working on the examined 120 construction sites (during the testing 120 scaffoldings), 573 employees took part in the study (573 surveys were carried out among people working at the construction site of the examined scaffoldings).
Comment: Line 261-262 - It is not necessary to repeat the consumption equivalence data, in parentheses. It has already been specified above.
Answer: Thank you very much for your comment. We removed this text.
Comment: Paragraph 269-275 - You must cite all the sources you are referencing (I think there are 7 references in this paragraph). Please be clearer in your explanations in the methodology section.
Answer: We added the follow references:
[63] Mishra, P.; Pandey, C.; Singh, U.; Gupta, A.; Sahu, C.; Keshri, A. Desciptive statistics and mornality tests for statistical data, Ann. Card. Anaesth. 2019, 22(1), 67-72, https://doi.org/10.4103/aca.ACA_157_18.
[64] Sijtsma, K.; Emons, W. Nonparametric statistical methods, International Encyclopedia of Education (Third Edition), 2010, 347-353, https://doi.org/10.1016/B978-0-08-044894-7.01353-1.
[65] Kelley, K. Confidence intervals for standardized effect sizes: theory, application, and implementation, J. Stat. Soft. 2007¸20(8), https://doi.org/10.18637/jss.v020.i08.
[66] Peng, C.-Y.; Lee, K.; Ingersoll, G. An introduction to logistic regression analysis and reporting. Int. J.Educ.Res. 2010, 96(1)¸ 3-14, https://doi.org/10.1080/00220670209598786.
[67] López-Martínez, A.; Exteve-Zarazaga, R.; Ramirez-Maestre, C. Perceived social support and coping responses are independent variables explaining pain adjustment among chronic pain patients. J. Pain. 2008, 9(4), 373-379, https://doi.org/10.1016/j.jpain.2007.12.002.
[68] Wang, H.; Peng, J.; Wang, B.; Lu, X.; Zheng, J.; Wand, K. Tu, X.; Feng, C. Inconsistency between univariate and multiple logistic regressions. Shanghai Archives of Psychiatry 2017, 25(2), 124-128, https://doi.org/ 10.11919/j.issn.1002-0829.217031.
[69] Mayo, D. Statistical inference as severe testing 2018, ISBN: 9781107286184, https://doi.org/10.1017/9781107286184.
Comment: Research results 4.1. Analysis of occupational accidents Line 332 - Table 2 indicates very interesting information. It should take two lines and explain the meaning of their corresponding values to each column. This paragraph should be incorporated before table 2.
Answer: Thank you very much for your comment. We rewrote table 2.
Comment: Analysis of survey data Line 350 - It is indicated that 573 surveys have been carried out. Has a survey been carried out on the 219 people injured in a work-related accident or on any of them?
Answer: As described in section 3 (Methodology of research) two independent datasets have been used. We did not survey conduct surveys in the first data source (accident documentation). It was impossible, because the first source of data concerns archival data.
Comment: Evaluator's comment - The results offered are based on a sample of 120 inspected works and 573 people. It could be very interesting to elaborate some results based on the type of construction: small, medium or large works. Or even, based on the type of company: small, medium or large company.
Keep in mind that you have a large number of results and may get different interpretations based on observation points. It is very important to be able to establish preventive actions to improve the problem of alcohol consumption. But I think your results are too generic and do not show the reality of the problem.
Answer: Thank you very much for your evaluator's comment. We are aware that collected data allow to conduct multiple analyzes. We analyze the collected data for various factors, such as: causes of accidents, assessment of the state of threat of working on construction scaffolding, etc. However, in this paper we focus on an issue that has not been discussed - i.e. alcohol. To date, there has not been any similar research carried out in Poland, and there is currently no information about alcohol consumption on Polish construction sites. Therefore, this paper is an attempt to fill this research gap.
Of course we have plan to do more analysis, among others: impact of the size of the scaffolding and enterprise on accidents or effect of smoking on behaviour, job performance and workplace safety with the use of scaffoldings.
Comment: Line 391 - In the description of table 4 it is better that you add the following: "Table 4. The number of people consuming alcohol (Yes), and the number of abstainers (No), with regards to their age [own elaboration].” Or provide a clarification or explanation.
Answer: Thank you very much for your comment. We added the proposed phrases.
Comment: Line 403 - Surely many photographs have been taken in each of the inspections carried out for data collection. However, the photographs shown only show the existence of beer cans. However, the photographs show the accumulation of debris as an element of risk. It is more interesting to show the types of buildings and scaffolds that have been inspected. It would have been better to provide a photograph with some workers drinking a can of beer. The faces in the photo can be blurred. I consider that the photographs are not representative. I should add some photos. If you consider the size of the photos to be inconvenient, you could reduce it. 8 photos could be placed on the page at a legible size.
Answer: Thank you very much for your comment. We added more information about tested 120 façade scaffoldings and added new photos as follow:
120 façade scaffoldings were tested. The façade scaffoldings are divided into 4 groups with regards to the surface area: 30-300 m2 (55 scaffolding), 300-600 m2 (28 scaffolding), 600-900 m2 (25 scaffolding) and 900-1,500 m2 (12 scaffolding). Photos 1-4 shows examples of tested scaffoldings with regards to the surface area.
Photo 1. Example of façade scaffold with an area of 30-300 m2 [authors' archive].
|
Photo 2. Example of façade scaffold with an area of 300-600 m2 [authors' archive].
|
Photo 3. Example of façade scaffold with an area of 600-900 m2 [authors' archive]. |
Photo 4. Example of façade scaffold with an area of 900-1,500 m2 [authors' archive].
|
You wrote that "It would have been better to provide a photograph with some workers drinking a can of beer"
We agree with you, but it was hard to do.
Firstly, workers knew that we conducted scientific research and their behaviour was close to “perfectly”.
Secondly, while examining the scaffolding and conducting a survey, we did not register situation that some workers drinking a can of beer.
Furthermore, unfortunately, but the construction workers consume alcohol in hard-to-reach places on the construction site (in hiding) and we did not have a chance to register it.
Comment: Summary of results This section shows a brief summary of the results obtained in the investigation. These are very interesting facts. However, on the one hand, in order to use this information, it is necessary to establish the type of accidents and the injuries or injuries they have caused (consider the need to incorporate this information). And, on the other hand, if it can be compared with other investigations in order to determine the degree of danger that working in a job with scaffolding has with respect to other types of jobs (this part could be referenced with the appropriate bibliography).
Answer: Thank you very much for your comment.
Answer to the first comment (necessary to establish the type of accidents): the type of accident was discussed in section 4.1 - especially on figure 5. Based on this information it was possible to declare that: "analysis of the content of alcohol in blood with regards to the effect of an accident indicates that as the amount of alcohol in the blood rises, the severity of the accident increases. A high alcohol content contributes to the death of a construction worker".
Answer to the second comment (it is necessary to establish the injuries or injuries they have caused): information about the injuries or injuries they have caused we could not use in this article. Generally an event that caused injury was a fall from scaffolding. As a result of a fall, employees suffered serious injuries. The type of injury is often not possible to identify at the time of the accident. In Poland, the final type of injury, or injuries, is determined within 6 months of the occurrence of the accident. Therefore, in post-accident reports, which are drawn up within 14 days of the occurrence of the accident, the information "indefinite injury as a result of falling from height" is very often included.
Answer to the third comment: we added new paragraphs with the appropriate bibliography as follow:
The place where an employee is working is constantly and dynamically changing [71]. With such dynamic environments, construction sites contain a significant quantity of unidentified or not well-assessed hazards that expose construction workers to additional safety risks during required operations [72]. Working on scaffolding carries a much higher risk of hazards than other types of construction works [73]. The main reason is the place of work itself - work at height. Working from height continues to be one of the major causes of fatality within construction industry [74].
[71] Yang, K.; Ahn, C.; Vuran, M.; Kim, H. Collective sensing of workers' gait patterns to identify fall hazards in construction Automat. Constr. 2017, 82, 166-178, https://doi.org/10.1016/j.autcon.2017.04.010.
[72] Albert, A.; Hallowell, M.; Kleiner, B.; Chen, A. Enhancing construction hazard recognition with high-fidelity augmented virtuality. J. Constr. Eng. Manag. 2014, 140(7), 4014024, https://doi.org/10.1061/(ASCE)CO.1943-7862.0000860.
[73] Szóstak, M. Analysis of occupational accidents in the construction industry with regards to selected time parameters. Open. Eng. 2019, 9(1), 312-320, https://doi.org/10.1515/eng-2019-0027.
[74] Sanchez, F.; Pelaez, G.; Alis, J. Occupational safety and health in construction: A review of applications and trends. Ind. Health 2017, 55, 210–218, https://doi.org/10.2486/indhealth.2016-0108.
Comment: Discussion Lines 443-447 - The sentences described are not the result of your investigation. Each of the sentences should be cited.
Answer: Thank you very much for your comment. These sentences has been moved to the Section 2 (Literature review) with appropriate bibliography.
Comment: Paragraph 448-454 - Each of the sentences must be cited, as it is not the result of your research. If the citation [39] refers to the entire paragraph, please, it is better that at the beginning of the paragraph describe who is the author of this research: "An investigation carried out by Liu et al [39], determined the following parameters ... "
Answer: Thank you very much for your comment. These sentences has been rewrote and removed to the Section 2 (Literature review) with appropriate bibliography as follow:
An investigation carried out by Liu et al. [44] determined that the average human body is able to burn 0.12‰ to 0.15‰ of alcohol per hour. The process of alcohol burning depends on many factors, among others, on sex, body weight, individual predispositions associated with metabolism, the amount and type of consumed food, and the state of health of the body. Small amounts of consumed alcohol are excreted with exhaled air and urine, only larger amounts attack the body and reach the brain and other organs. Excessive and prolonged alcohol consumption can lead to alcoholic liver damage (liver cirrhosis), myocardial damage and brain damage (ischemic stroke).
Comment: It is not understood why the discussion section makes references to other research. If citations are provided, it is to serve as a discussion towards the results of your research. You should incorporate data from your research that discusses with data from other researches. Please, if you do not have results you will have to move from line 437 to line 454 to the Literature Review section. Also, warn you that the quote [39] refers to China. You should consider expanding this information with quotes from closer countries in order to establish a proper basis for discussion.
Answer: Thank you very much for your comment. These sentences has been removed to the Section 2 (Literature review).
Comment: Line 455 - The expression "Therefore ..." determines a continuity in the content of the section. You cannot establish a continuity in the discussions when the previous two paragraphs are quotes and are not the result of your research.
Answer: Thank you very much for your comment. We rewrote this paragraph.
Comment: Lines 466-468 - The sentence "It should be ... ... differences." must be cited. and the sentence "The alcohol ... ... vary greatly." must be cited.
Answer: Thank you very much for your comment. We modified and updated references as follow:
[54] Caderbaum, A. Alcohol metabolism. Clin. Liver. Dis. 2012, 16(4), 667-685, https://doi.org/10.1016/j.cld.2012.08.002.
[55] Edenberg, H. The Genetics of alcohol metabolism: role of alcohol dehydrogenase and aldehyde dehydrogenase variants. Alcohol Res. Health, 2007, 30(1), 5-13.
Comment: Lines 470-472 - The following clarification is made: "Since there is no officially recorded data, results obtained from the data collection method could be biased. Findings might have been subject to selection bias, although the data was collected with the greatest care. " I recommend you delete this clarification. To do this, you will have to give an explanation in the methodology section about how the data was collected and the data selected in the surveys. Please note that this statement could lead to the article not being published. For this, there are different types of psychosocial standards that guarantee the veracity of the answers. That is why it is important that you explain how the survey was developed.
Answer: Thank you very much for your comment. We removed these sentences and added more detail in section 3.2 (Suryeys) about how the data was collected and the data selected in the surveys as follow:
Data collection method included questionnaires. The data obtained in this study were direct responses from individuals. As part of the research have been developed standardized protocols for data collection. Furthermore, all study personnel has been trained to conduct the research. It is well known that training of study personnel allows to minimize inter-observer variability [60]. The questions concerned private or sensitive topics, such as consume alcohol. Thus, self-reporting data may have been affected by an external bias caused by social desirability. The bias in this case can be referred to as social desirability bias [61]. Moreover, at the stage of validation, the data obtained from the questionnaire was analyzed using the methods of descriptive statistics in order to verify the variability of responses to individual questions. What is important, that the analysis of the collected data was performed after all the surveys had been completed. This approach was intended to reduce the interviewer's bias [62].
[60] Pannucci, C.; Wilkins, E. Identyfying and avoiding bias in research. Plast. Reconstr. Surg. 2011, 126(6), 619-625, https://doi.org/10.1097/PRS.0b013e3181de24bc.
[61] Althubaiti, A. Information bias in jealth research: definition, pitfalls, and adjustment methods. J. Multidiscip. Healthc., 2016, 9, 211-217, https://doi.org/10.2147/JMDH.S104807.
[62] Davis, R.; Couper, M. Janz, N. Caldwell, C. Resnicow, K. Interviewer effects in public health surveys. Health Educ. Res. 2010, 25(1), 14-26, https://doi.org/10.1093/her/cyp046.
Comment: Lines 480-481 - "It is worth noting here that in Poland, unlike other European countries..." I recommend that you do not make such an inappropriate reference. It is better that you make references to certain countries (quoting) or to several countries in general. For example: "It is worth noting that both in Poland and in other European countries ..."
Answer: Thank you very much for your comment. We agree with the Reviewer and we changed this sentence.
Comment: Conclusions The conclusions that are proposed do not refer to the results of the investigation nor do they propose possible solutions. From a general approach, a particular conclusion can be proposed regarding your research. The conclusions they contribute to this research are not relevant and should be rewritten. It has to be based on your own research, its results and the benefits it can bring to society.
Answer: Thank you very much for your comment. We rewrote this section and we hope that in the current version of the paper, we have taken into account all your critical remarks. Besides the resubmission of the manuscript we are also enclosing the paper in the track changes which may be handy for the reviewers to spot the areas of our improvements in this section.
Comment: The statement (line 505): "... should be allowed to carry out sobriety checks of employees." What does the law say about it? This is more of a discussion than a conclusion. Line 508 - there is an appointment; the [40]. No bibliographic references are made in the conclusions section. They are conclusions from the results of your research.
Answer: Thank you very much for your comment. We agree with the Reviewer and removed this paragraph to the Section 5 (Discussion) with appropriate bibliography:
[75] Act of 26 October 1982 on upbringing in sobriety and counteracting alcoholism (Law Gazette, No. 35, it. 230 as amended) (in Polish), 1982. Available online: http://prawo.sejm.gov.pl/isap.nsf/DocDetails.xsp?id=WDU19820350230 (accessed on 23 April 2020).
Comment: References In references, it is important that you add the DOI information (if any) for each item. For example, the first reference:
[1] Miller, T.R .; Zaloshnja, E .; Spicer, R.S. Effectiveness and benefit-cost of peer-based workplace substance abuse prevention coupled with random testing. Accid. Anal. Prev. 2007, 39, 565–573. https://doi.org/10.1016/j.aap.2006.10.001
The percentage of current references is good, but remember that the new references that you will have to add must be current as well.
Answer: Thank you very much for your comment. We added DOI information.

Reviewer 2 Report
Employees in construction industry are influenced by weather (sun-rain), period (summer – winter). Is there a different withdrawal of alcohol from the human body when the environment conditions are changed? Did the authors take this into account different environment conditions in the calculation – any coefficient? Did they consider this factor at all?
Supplement the text with a map of the sites in Poland where the research was realised. Can be this map be considered as another factor for asessment?
Figures 2 and 4 - I recommend to create a bar graph, or another type of graph.
Author Response
Dear Reviewer,
Thank you very much for your review and your critical comment, which allow to do our article better. We apologize that the previous version of our paper did not meet your expectations. We hope that in the current version of the paper, we have taken into account all your critical remarks. We also hope, that the current version meets your expectations.
Below we present the point-by-point response to received reviews and the answers to the most important comments and suggestions for Authors.
Comment: Employees in construction industry are influenced by weather (sun-rain), period (summer – winter). Is there a different withdrawal of alcohol from the human body when the environment conditions are changed? Did the authors take this into account different environment conditions in the calculation – any coefficient? Did they consider this factor at all?
Answer: Thank you very much for your comment. We agree with the reviewer's comments, that the changing environment conditions (especially weather conditions) an influence on withdrawal of alcohol from the human body.
Unfortunately in this research we did not take different environment conditions in the calculation.
The comments indicated by the Reviewer about "the weather (sun-rain), period (summer – winter) are the subject of detailed analyzes currently conducted by the Authors. In subsequent works, the Reviewer suggestion will be discussed in detail and supported by the analyzes.
Comment: Supplement the text with a map of the sites in Poland where the research was realised. Can be this map be considered as another factor for asessment?
Answer: We completed text with a map of the sites in Poland where the research was realised as follow:
The figure 1 presents the map of the sites in Poland where the research was realized.
Figure 1. The map of the sites in Poland where the research was realized [own elaboration].
Comment: Figures 2 and 4 - I recommend to create a bar graph, or another type of graph.
Answer: We agree with the reviewer's comments. At the present version of this paper we improved it.
Figure 4. Age structure of injured people who were under the influence of alcohol based on Control Protocols [own elaboration].
Figure 6. The age structure of the respondents [own elaboration].

Reviewer 3 Report
This manuscript investigates an interesting topic, but I have serious concerns related to the research methods and data sources that have been used. First, the accident data source is limited to 219 number of accidents of which only 22 are related to alcohol abuse. This is very limited to make any meaningful conclusions. Second, using the survey data for alcohol abuse is subjected to cognitive biases and I have doubts regarding the reliability of findings. Third, the research method is simply reporting descriptive data without any robust statistical analysis. Forth, figures 1 and 2 (lines 406-410) do not show any significant findings, people might have drink beer after work or maybe someone just throw those cans there. Finally, the manuscript needs to be proofread, the figure numbers are not correct, and text needs to be edited.
Author Response
Dear Reviewer,
Thank you very much for your review and your critical comment, which allow to do our article better. We apologize that the previous version of our paper did not meet your expectations. We hope that in the current version of the paper, we have taken into account all your critical remarks. We also hope, that the current version meets your expectations.
Below we present the point-by-point response to received reviews and the answers to the most important comments and suggestions for Authors.
Comment: This manuscript investigates an interesting topic, but I have serious concerns related to the research methods and data sources that have been used.
First, the accident data source is limited to 219 number of accidents of which only 22 are related to alcohol abuse. This is very limited to make any meaningful conclusions.
Answer: Thank you very much for your comment. We agree with the Reviewer, but as we wrote, that: “The cause related to alcohol consumption occurred in 38 injured people”. However, only “in 22 Control Protocols, labour inspectors determined the exact value of the alcohol content in the blood of the injured person”. Unfortunately the accident documentation (inspection reports) was prepared by inspectors of the National Labour Inspectorate and we were not able to interfere or supplement this archival data.
We are aware that this is a small research sample and the obtained results do not apply to the entire population of employees. we are aware that this is a small research sample and the obtained results do not apply to the entire population of employees. However, we hope to expand the database for new data in the near future.
Comment: Second, using the survey data for alcohol abuse is subjected to cognitive biases and I have doubts regarding the reliability of findings.
Answer: Thank you very much for your comment. The data obtained in this study were direct responses from individuals. We are aware that using the survey data for alcohol abuse is subjected to cognitive biases. Since there is no officially recorded data, results obtained from the data collection method could be biased. Findings might have been subject to selection bias, although the data was collected with the greatest care.
Data collection method included questionnaires. The data obtained in this study were direct responses from individuals. As part of the research have been developed standardized protocols for data collection. Furthermore, all study personnel has been trained to conduct the research. It is well known that training of study personnel allows to minimize inter-observer variability]. The questions concerned private or sensitive topics, such as consume alcohol. Thus, self-reporting data may have been affected by an external bias caused by social desirability. The bias in this case can be referred to as social desirability bias]. Moreover, at the stage of validation, the data obtained from the questionnaire was analyzed using the methods of descriptive statistics in order to verify the variability of responses to individual questions. What is important, that the analysis of the collected data was performed after all the surveys had been completed. This approach was intended to reduce the interviewer's bias.
Therefore we believe that the obtained results are reliable and true.
Comment: Third, the research method is simply reporting descriptive data without any robust statistical analysis.
Answer: Thank you very much for your comment. We agree with the Reviewer that in the article we used descriptive statistics mainly, but elements of statistical analysis were also included for the second data source (normality of distributions of continuous variables, the Shapiro–Wilk test, the logistic regression analyses). We believe that at this stage of the study used statistical tools are sufficient
Comment: Forth, figures 1 and 2 (lines 406-410) do not show any significant findings, people might have drink beer after work or maybe someone just throw those cans there.
Answer: Thank you very much for your comment. We agree with you, but it was hard to photograph with workers drinking a can of beer.
Firstly, workers knew that we conducted scientific research and their behaviour was close to “perfectly”.
Secondly, while examining the scaffolding and conducting a survey, we did not register situation that some workers drinking a can of beer.
Furthermore, unfortunately, but the construction workers consume alcohol in hard-to-reach places on the construction site (in hiding) and we did not have a chance to register it.
Comment: Finally, the manuscript needs to be proofread, the figure numbers are not correct, and text needs to be edited.
Answer: Thank you very much for your comment. We rewrote all sections of the article and we hope that in the current version of the paper, we have taken into account all 3 Reviewers critical remarks. Besides the resubmission of the manuscript we are also enclosing the paper in the track changes which may be handy for the reviewers to spot the areas of our improvements in this section.

Round 2
Reviewer 1 Report
Article Title
The title is correct and seems to clearly identify the purpose of the document.
Abstract
The abstract offers a very clear view of the research. Well-structured and concise. Important: Your abstract contains 318 words. The journal may limit the number of words in the abstract to 200. If so, reduce the number of words while maintaining the same structure and fundamental concepts, please.
Keywords
The keywords are correct. If the magazine tells you to reduce the number of concepts, I recommend that you eliminate the concepts of "Occupational Accidents" and "Drugs"
- Introduction
The changes made to the introduction have significantly improved the reasons for the investigation. The references are current, and the texts are duly cited. The structure offers a clear idea of the enormous problem that exists with the abusive consumption of alcohol, generically and on construction sites.
- Literature review
The changes made give a good idea of this problem. Different investigations are adequately cited. The state of the art is very well founded. In addition, the text structure has improved.
- Methodology of research
This section has improved a lot. The procedure for the investigation is much better understood now.
3.1. Accident documentation
Alcohol abuse is a very widespread problem in many European countries. The research carried out covers a very wide area of the country, according to the areas marked in figure 1. The photograph in figure 2 is important and significant, and it is also very successful.
3.2. Surveys
This section is clear and properly cited.
- Research results
4.1. Analysis of occupational accidents
This section is clear and properly cited.
4.2. Analysis of survey data
The photographs shown identify different types and sizes of scaffolds. This research is very important because the consequences of an abusive consumption of alcohol in jobs with scaffolds are or can be very high. I appreciate your comments. Do not forget that you should increase your research on the different types of jobs that exist in a construction site: excavation, foundation, structure, facade, roof, facilities and finishing elements.
I appreciate your comments.
This section is clear and properly cited.
4.3. Summary of results
This section is clear and properly cited.
- Discussion
Changes made have improved this section and offers a self-critical view based on research.
- Conclusions
Unfortunately, this section is very poor in the conclusions of the investigation.
There are paragraphs that have moved them to other more suitable sections, with the changes required in the first evaluation. However, the research has a lot of information from which a lot of conclusions can be drawn. Note that the objective of the research was to take data (two sources) and determine some results. But all research has a purpose. Please make an effort. You can make conclusions, based on the results obtained, regarding the data collection from the two sources, regarding the results, regarding the differences between the research areas (north, south, east, west; city-town). Your conclusions will be very relevant and necessary for the publication of this article.
Please, you have to give importance to the research you have done and offer the reader different circumstances and purposes of your research.
References
References are properly written. They are current in a high percentage and the incorporation of the DOI code is appreciated.
Author Response
Dear Reviewer,
Thank you very much for your review and your critical comment, which allow to do our article better. We apologize that the previous version of our paper did not meet your expectations. We hope that in the current version of the paper, we have taken into account all your critical remarks. We also hope, that the current version meets your expectations.
Below we present the point-by-point response to received reviews and the answers to the most important comments and suggestions for Authors.
Comment: Abstract. The abstract offers a very clear view of the research. Well-structured and concise. Important: Your abstract contains 318 words. The journal may limit the number of words in the abstract to 200. If so, reduce the number of words while maintaining the same structure and fundamental concepts, please.
Answer: Thank you very much for your comment. We have shortened the summary to the required 200 words.
Comment: Keywords. The keywords are correct. If the magazine tells you to reduce the number of concepts, I recommend that you eliminate the concepts of "Occupational Accidents" and "Drugs
Answer: Thank you very much for your comment. We reduced the number of keywords.
Comment: Conclusions. Unfortunately, this section is very poor in the conclusions of the investigation.
There are paragraphs that have moved them to other more suitable sections, with the changes required in the first evaluation. However, the research has a lot of information from which a lot of conclusions can be drawn. Note that the objective of the research was to take data (two sources) and determine some results. But all research has a purpose. Please make an effort. You can make conclusions, based on the results obtained, regarding the data collection from the two sources, regarding the results, regarding the differences between the research areas (north, south, east, west; city-town). Your conclusions will be very relevant and necessary for the publication of this article.
Please, you have to give importance to the research you have done and offer the reader different circumstances and purposes of your research.
Answer: Thank you very much for your comment. We added more conclusions as follow:
Data for the analysis was obtained from two sources. The archival post-accident documentation, which was the first source of data from 2008-2017, allowed to determine the most probable scenario of alcohol consumption by employees during work. The advantage of this studies was the 10-year period of data collection, which allowed to establish a certain trend in the accident situation. Unfortunately, the data were prepared by various inspectors of the National Labour Inspectorate and contained varying degrees of detail (from 38 pos-accident documentation - 16 protocols only state that ""injured person consumed alcohol" or "under the influence of alcohol"). Therefore, when planning this type of research, it should be remembered that we may be dealing with incomplete data when using archival data. On the other hand, surveys (the second source of data) required the researchers to proper plan and conduct of surveys. When preparing the questionnaire, it is important to remember that the questions should be logical, understandable to the respondents and not suggestive answers. It is also very important to properly train the study personnel. The advantage of this studies was the testing of nearly 40% of people working at workplaces with scaffoldings.
And
Comparative analysis of the results from the post-accident documentation and survey showed that the number of people consuming alcohol decreases with age but the number of alcohol-related accidents does not decrease. Moreover, the percentage of people consuming alcohol slightly changes with age.
The research was conducted in 5 research areas, i.e. in 5 provinces (voivodeships) of Poland (Dolnośląskie, Lubelskie, Łódzkie, Mazowieckie, Wielkopolskie). The research of post-accident documentation showed that most of the accidents related to alcohol took place in central Poland, i.e. in the Łódzkie Voivodeship (12 accidents, i.e. that every 4 accidents were related to alcohol). On the other hand, the surveys showed that most of the people consuming alcohol work on construction sites in the central and eastern part of Poland, i.e. in the Mazowieckie Voivodeship (59% of working people) and Lubelskie Voivodeship (57% of working people). Moreover, according to the answers provided by the respondents, the largest number of people who did not drink while working (i.e. abstinents) was in the Wielkopolskie Voivodeship (71%).

Reviewer 3 Report
Unfortunately, I still believe that shortcomings of this study are significant and cannot be published. Authors need to re think data collection and analysis approach.
Author Response
Dear Reviewer,
Thank you very much for your review and your critical comment.
Comment: Unfortunately, I still believe that shortcomings of this study are significant and cannot be published. Authors need to re think data collection and analysis approach.
Answer: Thank you very much for your comment, but we cannot agree with your comments.
This article is the result of the implementation the research project "Model of the assessment of risk of the occurrence of building catastrophes, accidents and dangerous events at workplaces with the use of scaffolding", financed by the National Centre for Research and Development within the framework of the Programme for Applied Research.
The research program was properly prepared and approved by the funder (The National Centre for Research and Development). Moreover, the report on the implementation of the project was positively assessed by the National Centre for Research and Development, and the project was considered to be fully completed.
In the revised version of this article, we took into account the received comments, i.e. we added more information about the data collection method, preparation of questionnaires. Moreover, in the article we used descriptive statistics mainly, but elements of statistical analysis were also included for the second data source (normality of distributions of continuous variables, the Shapiro–Wilk test, the logistic regression analyses). Furthermore, we rewrote all sections and improved the article according to the other 2 reviews.
We still hope, that the current version meets your expectations.
